# Diagnostic Performance of Biomarkers for Bladder Cancer Detection Suitable for Community and Primary Care Settings: A Systematic Review and Meta-Analysis

**DOI:** 10.3390/cancers15030709

**Published:** 2023-01-24

**Authors:** Evie Papavasiliou, Valerie A. Sills, Natalia Calanzani, Hannah Harrison, Claudia Snudden, Erica di Martino, Andy Cowan, Dawnya Behiyat, Rachel Boscott, Sapphire Tan, Jennifer Bovaird, Grant D. Stewart, Fiona M. Walter, Yin Zhou

**Affiliations:** 1The Primary Care Unit, Department of Public Health and Primary Care, University of Cambridge, Cambridge CB1 8RN, UK; 2Centre for Cancer Genetic Epidemiology, Department of Public Health and Primary Care, University of Cambridge, Cambridge CB2 0SR, UK; 3Division of Primary Care, Public Health & Palliative Care, Leeds Institute of Health Sciences, University of Leeds, Leeds LS2 3AA, UK; 4Patient & Public Representative c/o The Primary Care Unit, Department of Public Health and Primary Care, University of Cambridge, Cambridge CB1 8RN, UK; 5Department of Surgery, University of Cambridge, Cambridge Biomedical Campus, Cambridge CB2 0QQ, UK; 6Wolfson Institute of Population Health, Barts and The London School of Medicine and Dentistry Queen Mary University of London, London EC1M 6BQ, UK

**Keywords:** bladder cancer, early detection, biomarkers, diagnostic performance, primary care, community

## Abstract

**Simple Summary:**

Bladder cancer (BC) is one of the most common cancers worldwide. Early-stage diagnosis is associated with better survival rates and, as such, the timely referral of suspected cases is paramount. Urinary biomarkers have been developed to aid diagnosis, and are largely tested in patients who have been referred for further investigation. Evidence, however, on their diagnostic performance for both detecting and ruling out BC, especially in the general population, is limited. In this review, we systematically identified studies reporting on the diagnostic performance of biomarkers suitable for use in primary and community care settings. Three biomarkers, with relatively little difference in diagnostic performance between them, and some novel biomarkers were identified showing potential to be used as a triage tool in such settings. While promising, further validation studies in the general population are needed.

**Abstract:**

Evidence on the use of biomarkers to detect bladder cancer in the general population is scarce. This study aimed to systematically review evidence on the diagnostic performance of biomarkers which might be suitable for use in community and primary care settings [PROSPERO Registration: CRD42021258754]. Database searches on MEDLINE and EMBASE from January 2000 to May 2022 resulted in 4914 unique citations, 44 of which met inclusion criteria. Included studies reported on 112 biomarkers and combinations. Heterogeneity of designs, populations and outcomes allowed for the meta-analysis of three biomarkers identified in at least five studies (NMP-22, UroVysion, uCyt+). These three biomarkers showed similar discriminative ability (adjusted AUC estimates ranging from 0.650 to 0.707), although for NMP-22 and UroVysion there was significant unexplained heterogeneity between included studies. Narrative synthesis revealed the potential of these biomarkers for use in the general population based on their reported clinical utility, including effects on clinicians, patients, and the healthcare system. Finally, we identified some promising novel biomarkers and biomarker combinations (*N* < 3 studies for each biomarker/combination) with negative predictive values of ≥90%. These biomarkers have potential for use as a triage tool in community and primary care settings for reducing unnecessary specialist referrals. Despite promising emerging evidence, further validation studies in the general population are required at different stages within the diagnostic pathway.

## 1. Introduction

Bladder cancer is the tenth most commonly diagnosed cancer worldwide, with 573,000 new cases and 213,000 deaths in 2020, ranking 14th in terms of cancer-associated mortality [1]. About two-thirds of patients with bladder cancer present with haematuria, a cardinal symptom for urological tract cancers including bladder cancer [2]. However, only visible haematuria has a high positive predictive value for bladder cancer, with pooled incidence reported to be as high as 17–18% in some populations [3]. Patients presenting with non-visible haematuria and other urological symptoms, such as lower urinary tract symptoms, may cause diagnostic challenges. This is because these symptoms are common in the general population and are more likely to be due to benign causes rather than cancer [4]. Identifying tools to improve the diagnostic pathway may improve diagnostic timeliness, and therefore outcomes, for patients with bladder cancer.

Empirical evidence suggests that there is scope to improve timely diagnosis and reduce missed diagnostic opportunities in symptomatic patients who are subsequently diagnosed with bladder cancer [5]. The diagnostic pathway for bladder cancer involves a combination of investigations, from urine tests in the community and primary care to specialist investigations such as upper urinary tract imaging (including ultrasound and Computed Tomography (CT) scans), urine cytology and cystoscopy. The latter remains the gold standard for bladder cancer detection in patients investigated following haematuria [6,7]. Disadvantages of these tests, such as the poor sensitivity of ultrasounds, radiation exposure associated with CT, and the invasiveness of cystoscopy, can limit their use in the general population [8]. There is, therefore, an urgent need to identify new approaches for improving risk stratification of symptomatic patients to improve the early detection of bladder cancer and reduce the burden of unnecessary investigations for patients.

Urinary biomarkers have been developed to aid detection and early diagnosis of urinary tract cancers including bladder cancer [9]. These have largely been tested in patients presenting with symptoms suggestive of cancer and referred for further investigation, and who are therefore at a higher-than-average risk of having an undetected cancer. Several reviews have been conducted focusing on individual biomarkers, biomarker panels or certain biomarker categories (e.g., proteins) seeking to describe and explore their diagnostic performance and clinical utility for bladder cancer detection [10,11,12,13]. These findings demonstrate that urinary biomarkers have the potential to improve current diagnostic strategies. However, it is not clear exactly when and how they should be used along the diagnostic pathway. 

This review aimed to update the evidence on existing biomarkers for bladder cancer detection, their diagnostic performance across different population groups, and explore their clinical utility in the populations and settings studied. The main focus was to identify novel biomarkers for bladder cancer detection that might be suitable for use in the general population in community and primary care settings, often the first point of contact for patients in the healthcare system, hereafter referred to jointly as community settings. For the purpose of this review, general population refers to patients with any baseline risk at the point of presentation, in contrast with patients referred to specialist settings, who are already at a higher risk of cancer. 

## 2. Methods

This review has been conducted in line with the guidance provided in the Cochrane Handbook for Systematic Reviews of Interventions [14] and reported following the Preferred Reporting Items for Systematic Reviews and Meta-Analysis of Diagnostic Test Accuracy Studies (PRISMA-DTA) Guidelines [15] (Appendix A). The Review Protocol has been registered on PROSPERO (registration ID: CRD42021258754). 

### 2.1. Data Sources

MEDLINE and EMBASE were systematically searched for studies reporting on primary data published in peer-reviewed journals between 1 January 2000 and 24 May 2022. Based on initial pilot searches and findings from previous reviews showing that most studies on novel biomarkers have been published since 2005, 2000 was set as the start date/year of database searches. The search strategy was developed in consultation with an experienced subject librarian. Data sources were supplemented by hand-searching of reference lists of included studies. 

### 2.2. Inclusion & Exclusion Criteria 

A comprehensive list of inclusion and exclusion criteria to guide searches and study selection (Appendix A) was developed using the PICOS Framework (Population, Intervention, Comparators/Context, Outcomes and Study type) [16]. Any studies i. involving adult patients presenting with clinical signs or symptoms suggestive of bladder cancer, undergoing evaluation but not having received a diagnosis; ii. comprising at least 50 bladder cancer and 50 non-bladder cancer patients; iii. recruiting participants through any healthcare system and setting; iv. reporting on at least one measure of diagnostic performance of biomarkers (either individual, multiple/panels or combinations), were included. We were interested in any biomarker feasible to use in community and primary care settings, i.e., identified in non-invasive samples such as blood (serum or plasma), urine, feces, saliva or breath.

This review was informed by the CanTest Framework, a 5-phase translational pathway for diagnostic tests, from new test discovery to health system implementation in low-prevalence populations [17]. Only studies providing measures of diagnostic accuracy beyond discovery/development (that is, Phase 2 and beyond of this framework) were included (Appendix A). As papers were set in different healthcare systems and settings, a diagram was developed to guide decisions on inclusion (adapted from Olesen et al. 2009) [18], using a cut-off point/boundary in the diagnostic pathway of a typical cancer patient. Studies were included up to the point where patients were referred for a first specialist visit but not diagnosed (see Figure 1). 

### 2.3. Search Strategy 

The search strategy (Appendix A) included a list of key words and database specific subject headings (MeSH or Emtree) relating to each of the main target domains: biomarkers, performance measures, early diagnosis, and bladder cancer, combined and tailored to the relevant database. Searches were limited to include only outputs published from January 2000 onwards. No restrictions on language or methodological design were applied. 

### 2.4. Study Selection 

Following deduplication in EndNote 20 (Clarivate Analytics), unique citations were imported to Rayyan—Intelligent Systematic Review software [19]. Title and abstracts were independently screened against inclusion/exclusion criteria by two reviewers (any two of NC/CS/VS, and EP). Full texts of potentially eligible papers identified during title and abstract screening were independently assessed by EP and one other reviewer (VS/AC/CS/EdM/RB/DB/ST/HH/YZ). Any disagreements were resolved through discussion and consensus. 

### 2.5. Data Extraction 

Data extraction was performed using Microsoft Excel 2015. A data extraction template was developed to record information on study details, validation overview, biomarker characteristics, performance measures including measures for comparison to urine cytology (the standard non-invasive urological evaluation to diagnose bladder cancer), and suitability for community use. When studies reported on different phases of biomarker development, data were extracted only for eligible phases (i.e., biomarkers and measures beyond the discovery phase). 

Data extraction was piloted for 10% of the included studies (EP and DB/RB) and the data extraction template was revised accordingly to ensure consistency and accuracy of the information extracted. Data extraction for the remaining 90% of studies was carried out independently by two reviewers (EP and VS/AC/CS/EdM/RB/DB/ST/HH/YZ).

### 2.6. Quality Assessment 

QUADAS-2, a tool designed to assess the quality of primary diagnostic studies, was used to assess the risk of bias and applicability of included studies [20]. The tool covers four domains: i. patient selection; ii. index test; iii. reference standards; iv. flow and timing. Each domain comprises a series of signaling questions, aimed at identifying areas of potential bias or concerns over applicability, rated as “high”, “low” or “unclear”. Quality assessment was performed independently by two reviewers (EP and VS/AC/CS/EdM/RB/DB/ST/HH/YZ). Quality assessment ratings were compared, and any inconsistencies observed were resolved through discussion and consensus. 

### 2.7. Data Analysis 

Forest plots of accuracy measures (sensitivity and specificity) were produced for all biomarkers investigated in three studies or more, using the mada package in R [21]. For the biomarkers with their performance reported in five or more studies, a meta-analysis was conducted to calculate the Hierarchal Summary Receiver Operating Curve (HSROC) using the bivariate random-effects approach developed by Reitsma et al. 2005 [22] with a linear mixed model [23]. 

Narrative synthesis, selected for its potential to assess and synthesize heterogeneous and complex evidence in a rigorous and replicable way [24,25,26,27], was also performed for the meta-analysis biomarkers. Reported authors conclusions/recommendations were deductively synthesized, being thematically aligned to effects on patients and clinicians (focusing on acceptability, benefits, and harms), and effects on health care systems (focusing on referral patterns and costs), adapted from Phases 3 to 5 from the CanTest Framework (Appendix A). Narrative synthesis was also used to assess and synthesize diagnostic performance of biomarkers examined in fewer than three studies but with a reported high negative predictive value (NPV ≥ 90.0%). This additional analysis aimed to identify novel biomarkers for bladder cancer detection that might be suitable for use in the general population, as a high NPV can provide reassurance that cancer is unlikely to be present [28,29]. 

## 3. Results

### 3.1. Selection Process 

Electronic database searches retrieved 6638 records, of which, following deduplication, title and abstract screening, 98 were assessed in full text for eligibility. Fifty-five records were excluded for either being outside the focus of the review (*N* = 8) or reporting on: (i). discovery-only findings (*N* = 8); (ii). numbers of cancer/non-cancer cases that were inadequate, unclear, or not possible to calculate (*N* = 24); (iii). populations already diagnosed with bladder cancer or under surveillance for bladder cancer recurrence (*N* = 14). One study was excluded due to integrity concerns (*N* = 1). One additional record was added, following the manual searching of reference lists of included studies, leading to a final sample of 44 studies included in the review (Figure 2). No studies were excluded during quality assessment. 

### 3.2. Study Characteristics

Included studies (*N* = 44 publications) originated from all continents (Table 1), predominantly Europe (*N* = 17) including Germany (*N* = 5), the UK (*N* = 4), Spain (*N* = 2), the Netherlands (*N* = 2), Belgium (*N* = 1), Denmark (*N* = 1), France (*N* = 1), and Sweden/Spain/Netherlands (*N* = 1). Ten studies (*N* = 10) originated from Africa, all of which were conducted in Egypt. Six studies (*N* = 6) were carried out in Asia, including five in China (*N* = 5) and one (*N* = 1) in Pakistan. There were four studies (*N* = 4) originating from the USA and two studies (*N* = 2) from Australia and/or New Zealand. Finally, for five studies (*N* = 5), information on country of research was not available. Studies were largely prospective (*N* = 34) and/or single-centered (*N* = 24), with different study designs including cohort (*N* = 15), trials (*N* = 3), cross-sectional (*N* = 2), case–control (*N* = 2), observational (*N* = 2), and evaluation (*N* = 1) studies. For nineteen studies (*N* = 19), there was no information on study design. The most common recruitment setting was in the hospital (*N* = 29) whereas four studies (*N* = 4) recruited from more than one setting (i.e., hospital and community). No information on recruitment setting was available for eleven studies (*N* = 11).

### 3.3. Risk of Bias 

Potential sources of bias were identified in all four domains of QUADAS-2 during quality assessment. Flow and timing was the domain in which most studies were assessed as being at high risk (*N* = 33) for failing to include all recruited patients in their analysis or not specifying whether there was an appropriate interval between index test(s) (the diagnostic test(s) that is/are evaluated against the reference standard) and reference standards (the best available method of determining whether people have a condition). Key sources of bias identified in studies classified as high risk for *index test* (*N* = 30) *and patient selection* (*N* = 22) included failing to pre-define test thresholds and failing to avoid inappropriate exclusions or not specifying the type of sampling employed. In terms of *reference standard*, the risk of bias was assessed as being unclear (*N* = 25) in most studies due to no information being provided on whether the results of the reference standard were interpreted without knowledge of the results of the index test. Concerns over applicability were not identified for any of the included studies (Appendix A). 

### 3.4. Population Characteristics 

Included studies (*N* = 44) reported on 28,527 participants including 9780 patients with cancer and 18,747 non-cancer patients (Table 1). Based on information from studies in which gender (*N* = 34) and age (*N* = 34) were reported, participants were predominantly male (63%), ranging from 18 to 110 years of age, with two studies including minors, one aged 14 and one 15 included as outliers in large samples. Of the 18,747 non-cancer patients, 9611 were further specified to include 2285 normal/healthy patients and 7326 with non-malignant or pre-malignant conditions such as cystitis, urolithiasis, dysuria, urethral stricture, and prostate hyperplasia. No such information was available for the remaining 9136 non-cancer patients. Clinical signs and/or symptoms at first presentation included haematuria (either visible or non-visible) reported in twenty-seven (*N* = 27) studies, and non-malignant or pre-malignant conditions reported in twenty-three (*N* = 23) studies. Finally, risks factors identified included smoking (*N* = 13), ethnicity/race (*N* = 4) and schistosomiasis, as well as an acute and chronic parasitic disease associated with bladder cancer (*N* = 9), reported in eighteen (*N* = 18) studies. Four (*N* = 4) studies reported on non-cancer patients with benign bladder tumors, a history of bladder cancer or bladder cancer diagnosis (included as outliers in large samples according to the set inclusion/exclusion criteria), these being considered as risk factors to developing bladder cancer. 

### 3.5. Biomarker Characteristics 

Included studies (*N* = 44) reported on 112 biomarkers (37 individual, 34 multiple/panels and 41 combinations) including biomarker/s and cytology OR biomarker/s and base models OR biomarkers/s and imaging (Table 2). Ninety-six of them (*N* = 96) were reported in only one study. In terms of category, 52 biomarkers were classified as proteins (including single proteins, combinations of proteins and combinations of proteins with prediction models, cytology, and other tests), 36 as DNAs and 18 as mRNAs (all following the same pattern as in proteins). There were also nine biomarkers combining proteins and mRNAs and six biomarkers combining proteins and DNAs. The discrepancy in the total number of biomarkers per category (*N* = 121) and the total number of biomarkers reported (*N* = 112) is due to different biomarker categories pertaining to the same biomarker being reported together (an example of this is Telomerase in Table 2). All biomarkers were sampled from urine—apart from one (CYFRA21-1), which was sampled from both urine and serum using a range of test platforms such as Enzyme-Linked Immunoassay (ELISA), Fluorescence In Situ Hybridization (FISH), lateral flow test, and different types of Polymerase Chain Reaction (PCR). 

### 3.6. Meta-Analyses

#### 3.6.1. Assessing Heterogeneity

Forest plots for sensitivity and specificity were produced for all biomarkers reported in three or more studies (NMP-22, UroVysion, uCyt+ (also referred to as ImmunoCyt+), BTAstat and FGFR3) (Appendix A). Variation in the accuracy measures (sensitivity and specificity) between studies may be explained by either the use of different thresholds/cut-off values to distinguish between a positive and negative result or heterogeneity in the study design (for example, differences in study setting or study design). The performance of the three biomarkers reported in five or more studies (NMP-22, UroVysion, uCyt+) was summarized by calculating the HSROC, which accounts for the variation in cut-off values (Figure 3). 

The twelve studies reporting accuracy measures for NMP22 use four different thresholds (ranging from 3.6–10 IU/mL), so the large range of sensitivities (0.27–0.90) and specificities (0.31–0.98) reported are unsurprising. However, most of the studies fall outside of the 95% confidence region of the HSROC model (Figure 3a), which suggests that other differences between the studies are also causing the variation in performance. This could be due to the diverse study populations in which NMP-22 was tested, including the following categories: country (five in Germany, four in the UK, one in Pakistan, and one in Australia/New Zealand; country of research in three studies was not provided), population age (with some studies enrolling participants of much wider age range than others), ethnicity, symptoms at presentation, and the extent to which risk factors such as smoking were addressed (Table 1). 

The ten studies reporting performance for UroVysion all used the same platform (FISH), which indicates that the range of reported sensitivity (0.38–0.96) may be due to other differences between the studies (the range of reported specificities is, however, relatively small (0.76–0.99)). This is supported by the meta-analysis, which finds that several of the UroVysion studies fall outside the 95% confidence region of the HSROC model (Figure 3b). This could be due to variation in the study population between studies, in terms of the country of research (four in Germany, two in China, two in the USA, one in Belgium; country of research in one study was not provided), age range of included population and symptoms at presentation (Table 1). 

The six studies reporting performance for uCyt+ all use the same threshold to determine positive test results (“at least one clear positive cell”), therefore, the relatively narrow range of sensitives (0.62–0.92) and specificities (0.72–0.81) reported is unsurprising. In the HSROC model, most of the studies fall in the 95% prediction region (Figure 3c). This suggests that the studies are relatively homogenous, likely carried out in similar settings and using comparable populations. This finding is supported by examination of the study characteristics (Table 1); all six studies were conducted in Europe (five in Germany and one in France), enrolled populations of similar age ranges (participants aged from 18 to 97) and patients had similar symptoms at first presentation. 

#### 3.6.2. Overall Performance and Sensitivity Analysis

The HSROC models for NMP-22, UroVysion and uCyt+ are compared in Figure 3d, and summary measures of discrimination (how well the test distinguishes between those with and without bladder cancer) are given. The estimated summary ROC curves show UroVysion has the best discrimination (AUC estimate: 0.876), slightly outperforming uCyt+ (AUC estimate: 0.827) and considerably outperforming than NMP-22 (AUC estimate: 0.748). However, the adjusted partial AUC estimates (accounting for the observed ranges of accuracy measures and normalized) have similar results for all three (0.650, 0.707 and 0.689 respectively). The overlap of the prediction regions (the 95% prediction region of the HSROC model estimates), further demonstrates that in this meta-analysis, no significant differences in discrimination are found between these three biomarkers.

The performance of NMP-22 was reported for two different platforms, ELISA (*N* = 10) and BladderChek (*N* = 3). In a sensitivity analysis (Appendix A), the performance of NMP-22 across both platforms was compared to the performance for ELISA only (there were insufficient BladderChek studies (*n* < 5) for a separate HSROC analysis). Although the three BladderChek studies all report relatively high specificities (0.81–0.96)—compared to the ELISA studies (0.34–0.88)—the reported sensitivities are variable (0.26–0.76); the HSROC analysis finds only minimal differences in discrimination between the ELISA-only studies and all NMP-22 studies (adjusted partial AUC of 0.701 and 0.689 respectively).

### 3.7. Narrative Synthesis 

#### 3.7.1. For Biomarkers Reported in Three Studies or More

In terms of acceptability, two biomarkers (NMP-22, using the BladderChek platform, and BTAstat) were highlighted as operator-independent, simple, and fast to analyze during patient visits, and, therefore, are suitable for use in the outpatient clinic [32,48,59]. However, acceptability could be compromised, as the diagnostic performance of all identified biomarkers was reported to be widely dependent on the severity of haematuria as quantified by urine dipstick analysis (NMP-22 ELISA, UroVysion, uCyt+) [65] the presence or absence of haematuria (BTAstat) [32] and in cases with acute clot retention (FGFR3) [67]. 

Considering the benefits, all biomarkers were reported to either improve bladder cancer detection or reduce burden on patients and health care providers when used in conjunction with urine cytology [56,59,63]. FGFR3 was observed to efficiently detect bladder cancer in patients with low grade tumors [50] whereas NMP-22 (unspecified test platform) and BTAstat were shown to outperform cytology in detecting G3 tumors, with the former also showing significantly higher detection rates for G1 and G2 tumors [58]. As for harmful results, increased rates of false positive results were reported for four out of six (including NMP-22 ELISA and BladderChek) biomarkers in the following cases: (i). in patients with urinary tract inflammation and/or infection (NMP-22 ELISA [49,55,57] and BTAstat [32,57]); (ii). haematuria (NMP-22 ELISA, uCyt+) [65], or microscopic haematuria (UroVysion) [69]; (iii). atypical urinary cytology and other risk factors such as older age or significant tobacco use (NMP-22 ELISA, uCyt+, UroVysion) [49,69]. 

In terms of referral patterns, UroVysion and FGFR3 were reported to help with triaging rapid referrals for haematuria [50,61] or to reduce the frequency (uCyt+) [56] or the number of unnecessary cystoscopy/cytology tests (NMP-22 ELISA) [57] in the healthcare systems in which they were assessed. Further details of their specific use within the diagnostic pathway, however, were not provided. Finally, when it comes to costs, BTAstat was reported to have the lowest estimated cost compared to NMP-22 (unspecified test platform), cytology, and flexible and rigid cystoscopy [58] as opposed to UroVysion [69,72] and uCyt+ that were reported to come at increased cost [64].

#### 3.7.2. For Biomarkers with High Negative Predictive Value

A summary of key findings pertaining to novel biomarkers that were investigated in fewer than three studies and which show potential for early detection of bladder cancer in the general population can be found in Table 3. Eight novel biomarkers/tests from all biomarker categories were purposively selected based on their reported high negative predictive value (NPV ≥ 90.0%), indicating their potential use in the general population for triaging patients for further investigations (see Table 3). 

## 4. Discussion

### 4.1. Summary of Main Findings 

This systematic review identified 44 studies reporting on 112 different biomarkers and combinations for bladder cancer detection. Most of the biomarkers identified were only reported in one study, with only three biomarkers (NMP-22, UroVysion and uCyt+) in a sufficient number of studies (*n* ≥ 5) to be included in the HSROC calculations. These biomarkers showed similar discriminative ability (adjusted AUC estimates ranging from 0.650 to 0.707). Narrative synthesis revealed the potential of some of these biomarkers for use in the general population, based on their reported clinical utility including diagnostic performance and effects on clinicians, patients, and the healthcare system. Finally, several novel biomarkers showed high negative predictive value indicating their potential for use in the general population presenting in community settings. 

### 4.2. Comparison with Existing Literature 

The calculated adjusted HSROC revealed small variations in discrimination across the three biomarkers included in the meta-analysis, all of which are well-established FDA-approved biomarkers, that have, either individually or comparatively, been explored in systematic reviews and/or meta-analyses before [13,73,74,75,76]. Heterogeneity beyond variation in adopted thresholds was also confirmed for these three biomarkers. This reported variation can be largely associated with a range of confounding factors mainly pertaining to the heterogeneity of included studies—this has also been identified as the main limitation in most meta-analyses conducted to date [74,75]. In addition to different probability thresholds, a series of methodological factors including the diversity in study designs and population samples, and the extent to which risk factors were also addressed in the study could potentially be influencing performance variation. Considering population samples in more detail, variation persisted not only in the numbers of participants enrolled but also in the composition of the cohorts studied (e.g., non-cancer patients ranging from healthy general population participants to hospital urology patients with or without benign pathology) and the range of symptoms reported at first presentation. There is, therefore, a risk of spectrum bias given the observed variation in the population samples in which the biomarkers were tested [77,78]. This risk was particularly evident for some of the biomarkers included in the meta-analysis such as NMP-22, which included studies that enrolled heterogeneous populations (particularly in terms of age, ethnicity, symptomatology at presentation, and risk factors). Therefore, extrapolating results to reaffirm the potential applicability of the reported biomarkers in the general population is challenging.

The complementary narrative synthesis aimed to ascertain such applicability, by investigating further the population, contextual and implementation factors. Findings indicated variation similar to that of the meta-analysis and are in line with evidence reported in previous reviews [12,79] and meta-analyses [13,80] about the potential of these biomarkers to effectively supplement cytology in bladder cancer detection or help with appropriate rapid referrals and reduce the number of unnecessary cystoscopies in the studied populations. However, certain barriers, such as diagnostic performance measures being affected by the degree of haematuria [32,65,67] or the inability of certain biomarkers to detect low grade tumors [38,58], were also identified, compromising their utility in the general population. Hence, the reported promising value of these biomarkers needs to be treated with caution.

A number of novel biomarkers (such as ADXBLADDER, CxBladder Triage and Xpert) or combinations (FGFR3 + TERT + HRAS + OXT1 + ONECUT2 + TWIST) were also reported to have high negative predictive values, indicating potential utility in community settings, as reassurance can be provided that cancer is an unlikely outcome [28,29]. This potential was also reaffirmed by the narrative synthesis. However, considering that these biomarkers were tested in either one or two studies only, validation studies in the general population are still required.

### 4.3. Strengths and Limitations 

Comprehensive literature searches were performed, strict eligibility criteria were set for study selection and explicit methods were employed for data extraction and data analysis. Heterogeneity, however, is the main limitation of this review, pertaining to various aspects of included studies such as study design, population samples, thresholds used and outcome measures. Such heterogeneity may distort meta-analysis and, as a result, reported results should be interpreted with caution. Another limitation that is relatively common to systematic reviews of biomarker performance is a lack of clarity or a low quality of reporting, with most included studies, when critically appraised, being assessed as either unclear or at high risk of bias in at least one domain. Finally, considering that narrative synthesis was based on original authors’ conclusions, an impartial assessment of those results reported as potentially promising might, to some extent, be compromised depending on how positives/negatives of each biomarker were portrayed by authors in different studies. 

### 4.4. Implication for Research and Practice 

This review identified biomarkers that could potentially be beneficial for use in community settings based on their diagnostic performance. Similar to conclusions from previous reviews [12,79] while there are promising results (particularly regarding high NPVs) for some biomarkers, additional validations are still needed in the community setting. Although an attempt to limit heterogeneity was made by only including patients with a suspicion of cancer at the point of recruitment, it is likely that levels of cancer risk vary even for this group across different studies. Furthermore, the included studies were evaluated as being at a higher risk of bias in more than one QUADAS-2 domain. Therefore, caution is warranted when generalizing performance results.

It is also important to consider the role of the biomarker within the cancer diagnostic pathway. In the general population, there is the need for a test to help better risk stratify patients with urological symptoms to facilitate clinical decision-making regarding the need for referral for subsequent cancer-specific investigations, similar to the use of fecal immunochemical testing for possible colorectal cancer [81,82,83]. We found no studies reporting the use of biomarkers for bladder cancer in this context. To assess the clinical utility of these biomarkers in the community, there is therefore a need to evaluate these biomarkers in the general population, at the pre-referral stage of the diagnostic process. 

Finally, evidence on the effects on patients, clinicians, and health care systems (reported in the narrative synthesis) was not widely reported across included studies. Therefore, despite years of biomarker development and testing, implementation and cost-effectiveness (Phases 3–5 in the CanTest framework) are still not often investigated [17]. No single biomarker with excellent diagnostic performance and corresponding implementation data was identified and the current findings do not allow for firm recommendations of any of the identified biomarkers for use in the general population. Novel biomarkers showing promising results need to be further evaluated, preferably prospectively, with consistency regarding populations, care settings and thresholds/cut-off points used.

## 5. Conclusions

In conclusion, findings from this systematic review suggest that certain biomarkers show potential to complement or improve current bladder cancer diagnostic strategies. Limited evidence on novel biomarkers shows that those with high NPV could be promising for use in community settings as a triage tool for appropriate and necessary referrals. More prospective studies are needed to further validate this promising evidence in the general population before establishing the exact place/role of these biomarkers within the diagnostic pathway.

## Figures and Tables

**Figure 1 cancers-15-00709-f001:**
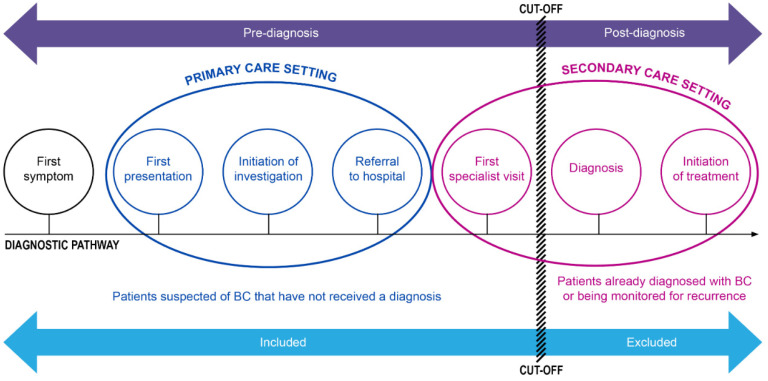
Cut-off/boundary in the diagnostic pathway for study inclusion.

**Figure 2 cancers-15-00709-f002:**
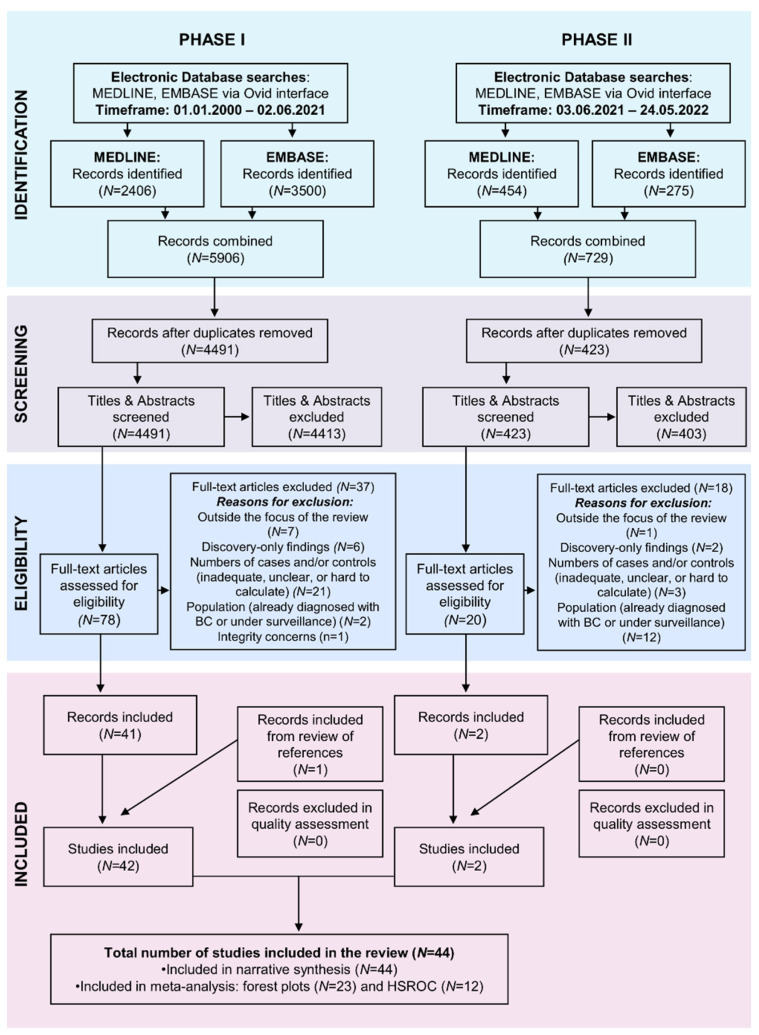
Preferred Reported Items for Systematic Reviews and Meta-Analyses (PRISMA) flow chart: Selection process for systematic review on the diagnostic performance of biomarkers for bladder cancer suitable for community settings.

**Figure 3 cancers-15-00709-f003:**
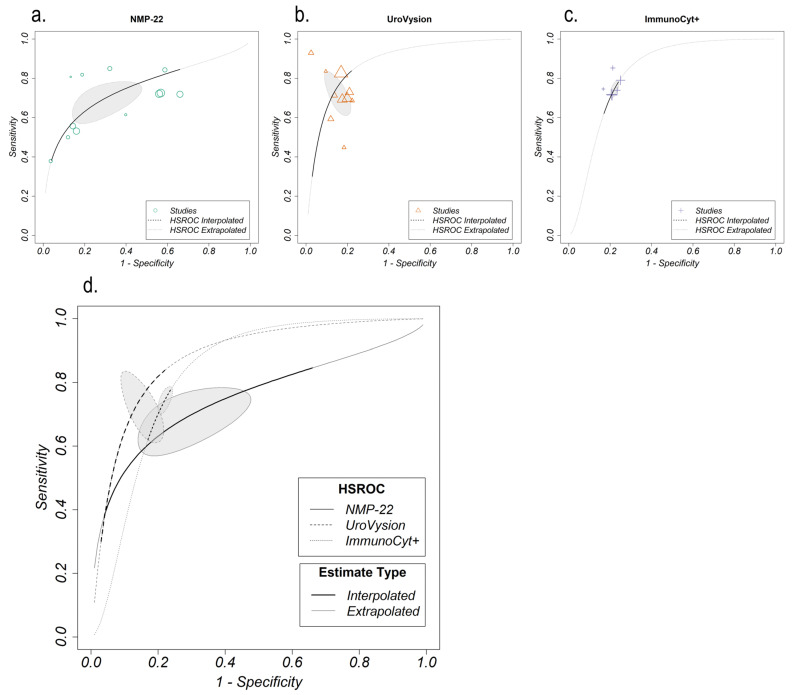
Hierarchical Summary of Receiving Operating Characteristics (HSROC) models for NMP-22 (**a**), UroVysion (**b**), uCyt+ (**c**) separately (**top**) and combined (**d**) (**bottom**).

**Table 1 cancers-15-00709-t001:** Study Characteristics—Context, Population & Intervention.

Context							Population													Intervention
Study	Country	Recruitment Timeline	Design			Clinical Setting: (Hospital (H), Community (Co), Unclear (U))	Cases	Controls			Age		Gender (M/F)	Risk Factors		Symptoms				Biomarker/Test
							Cases (C)	All (Con)	Non-Malignant (NM)	Healthy Controls (HC)	Range	Mean (SD) or Median ^1^		Smoking (*N*)	Other (*N*)	Haematuria		UTIs	Other	
																VH	NVH			
Attallah et al. 2015 [30]	Egypt	NA	Prospective	Single centre	NA	H	66	94	54	40	C (44–81) NM (45–70) HC (45–65)	C (60.6 ± 10.6) NM (58.6 ± 11.2) HC (56.2 ± 8.2)	106M:54F	-	-	x	-	-	Cystitis	EMA
																				NMP-52
																				EMA + NMP-52
Barbieri et al. 2011 [31]	USA	2001–2002	Retrospective	Multi-centre	Cohort	H&Co	72	1231	-	-	-	66 ± 13.6	738M:565F	474Y:829N	-	x	-	-		Base model (age, gender, smoker, race, haematuria) + NMP-22
																				Base model (age, gender, smoker, race, haematuria) + NMP-22 + Cytology
Bhuiyan et al. 2003 [32]	NA	NA	NA	NA	NA	U	70	163	-	-	-	-	-	-	-	-	-	-	-	Telomerase
																				BTAstat
																				NMP-22
																				Telomerase + Hb Dipstick
Critselis et al. 2019 [33]	Spain	2009–2012	Prospective	Single centre	Cross-sectional	H	179	85	85	0	-	69.4 ± 12.2	218M:46F	-	-	x		-	-	SPARC
Dahmcke et al. 2016 [34]	Denmark	2013–2015	Prospective	Single centre	Evaluation	H	99	376	-	-	C (26–91) Con (18–91)	C 69 Con 64	355M:120F	-	-	x	-	-	-	TERT, FGFR3 and SALL3, ONECUT2, CCNA1, BCL2, EOMES, and VIM
																				TERT
																				ONECUT2
																				VIM
																				SALL3
																				CCNA1
																				BCL2
																				EOMES
																				FGFR3
																				TERT + FGFR3
																				CCNA1 + ONECUT2 + BCL2 + EOMES + SALL3 + VIM
																				TERT + FGFR3 + CCNA1 + ONECUT2 + BCL2 + EOMES + SALL3 + VIM
																				TERT + FGFR3 + N + A1 + ONECUT2 + BCL2 + EOMES + SALL3
																				TERT + FGFR3 + CCNA1 + ONECUT2 + BCL2 + EOMES
																				TERT + FGFR3 + CCNA1 + ONECUT2 + BCL2
																				TERT + FGFR3 + CCNA1 + ONECUT2
																				TERT + FGFR3 + CCNA1
																				TERT + ONECUT2
Davidson et al. 2020 [35]	New Zealand	2018-2019	Prospective	Multi-centre	Cohort	H&Co	51	833	833	0	(14–97)	63.1 ± 16.2; median:65		71 Current: 293 Former: 520 Never	1 Previous cancer diagnosis; 1 Radiation therapy of pelvis	x	x	x	Cystitis; Upper tract stones; vascular Prostate; Anticoagulation; Renal disease; Primary amyloidosis	CxBladder Triage (CxbT)
																				CxbT + Imaging
Deininger et al. 2017 [36]	Germany	2006–2009	Retrospective	Single centre	Cohort	H	68	376	-	-	(18–93)	median:67	376M:68F	61 Current: 82 Former: 274 Never: 27 n/a	-	x		x	Benign prostatic hyperplasia; Prostate cancer	uCyt+ ^7^
Dudderidge et al. 2020 [37]	UK	2016–2017	Prospective	Multi-centre	Cohort	H	74	782	782	-	(54–73)	median:64	487M:369F	-	-	x	x	x	-	ADXBLADDER
Eissa et al. 2007 [38]	Egypt	NA	Prospective	Single centre	NA	H	200	115	85	30	C (37–78) NM (34–70) HC (30–78)	C (57.8 ± 10) NM (51 ± 11.8) HC (54.1 ± 9.7)	-	136Y:64N ^2^	85 Benign schistosomal urological lesions	-	-	-	-	RTA
																				hRT
																				hTERT
																				RTA + hRT
																				RTA + hTERT
																				hTR + hTERT
																				RTA + hRT + hTERT
																				HRT + Cytology
																				hRT + Cytology
																				RTA + hRT + Cytology
																				RTA + hTERT + Cytology
																				RTA + hRT + HTERT + Cytology
Eissa et al. 2007 [39]	Egypt	NA	Prospective	Single centre	NA	H	154	90	60	30	C (25–82) NM (25–79) HC (27–50)	C (57 ± 12) NM (51± 14) HC (45.36 ± 5.2)	164M:80F	-	60 Benign schistosomal urologic lesions	-	-	-	-	MMP-2
																				MMP-9
																				TIMP-2
																				MMP-2 + TIMP-2
																				MMP-9 + TIMP2
																				MMP zymography
																				MMP-2 + Cytology
																				MMP-9 + Cytology
																				TIMP-2 + Cytology
																				MMP-2+ TIMP-2 ratio + Cytology
																				MMP-9+ TIMP-2 ratio + Cytology
																				MMP zymography + Cytology
Eissa et al. 2007 [40]	Egypt	NA	Prospective	Single centre	NA	H	120	109	54	55	C (40–80) NM (17–80) HC (17–50)	C (55.0 ± 7.76) NM (49.8 ± 15.75) HC (21.8 ± 8.47)	-	-	-	-	-	-	Cystitis; Ureteral stone; Renal stone; Nephritis; Combined disorders	TGF-β1
																				VEGF
																				TGF-β1 + VEGF
																				TGF-β1 + VEGF + Cytology
																				TGF-β1 + Cytology
																				VEGF + Cytology
Eissa et al. 2009 [41]	Egypt	NA	Prospective	Single centre	Case-control	H	240	218	108	110	182 <45: 276 >45	-	278M:180F	217Y:241N	70 Schistosomal dysplasia; 28 schistosomal cystitis	-	-	-	benign prostatic hyperplasia	VEGF
																				bFGF
																				ANG
																				HGF
Eissa et al. 2010 [42]	Egypt	NA	NA	NA	NA	U	166	212	112	100	C (26–83) NM (18–75) HC (25–60)	C (60 ± 11) NM (48 ± 15) HC (49 ± 10)	-	-	-	-	-	-	-	HYAL-1 Qual
																				HYAL-1 Semi-quant
																				Survivin Qual
																				Survivin Semi-quant
																				HYAL-1 + Survivin Qual
																				HYAL-1 + Survivin Semi-quant
																				HYAL-1 Qual + Cytology
																				HYAL-1 Semi-quant + Cytology
																				Survivin Qual + Cytology
																				Survivin Semi-quant + Cytology
																				HYAL-1 Qual + Survivin Qual + Cytology
																				HYAL-1 Semi-quant + Survivin Semi-quant + Cytology
Eissa et al. 2011 [43]	Egypt	NA	Prospective	Single centre	NA	H	132	108	60	48	C (30–78) NM (25–72) HC (25–45)	C (56.60 ± 9.60) NM (51.30 ± 14.0) HC (29.20 ± 10.60)	-	-	-	-	-	-	-	FN
																				RTA
																				CK20
																				CK20 + RTA
																				CK20 + FN + RTA
																				FN + Cytology
																				CK20 + Cytology
																				RTA+ Cytology
																				FN+ CK20+ Cytology
																				CK20 + RTA + Cytology
																				FN + CK20 + RTA + Cytology
Eissa et al. 2012 [44]	Egypt	NA	Prospective	Single centre	NA	H	100	116	65	51	C (37–78) NM (31–75) HC (29–67)	C (57.40 ± 9.60) NM (52.6 ± 13.8) HC (46.9 ± 9.87)	-	64Y:36N^2^	65 Schistosomiasis positive	-	-	-	Cystitis; Ureteral stone; Renal stone; Nephritis; Combined disorders	RAR-B2
																				HYAL-1
																				RAR-β2 + HYAL-1
																				RAR-β2 + Cytology
																				HYAL-1 + Cytology
																				RAR-β2 + HYAL-1 + Cytology
Eissa et al. 2014 [45]	Egypt	2011-2012	Prospective	Single centre	NA	H	211	133	71	62	C (25–79) NM (25–66) HC (29–76)	C (52 ± 10) NM (50 ± 8) HC (51 ± 10)	252M:91F 1 Unknown	101Y:243N	193 Schistosomiasis positive	-	-	-	-	HURP
																				HURP + Cytology
Eissa et al. 2014 [46]	Egypt	2011–2012	Prospective	Single centre	NA	H	50	50	25	25	C (37–79) NM (25–68) HC (28–79)	C (59.8 ± 9.3) NM (52.88 ± 9.4) HC (45.44 ± 11.9)	78M:22F	38Y:62N	Schistosomiasis dysplastic lesions	x		-	Renal stone; Urethral stricture; Chronic irritative symptoms	HURP
																				HURP + Cytology
Fu et al. 2018 [47]	China	2016	Prospective	Single centre	NA	H	152	82	82	0	-	C (63 ± 13) Con (64 ± 13)	179M:55F	-	-	-	-	x	Cystitis; urolithiasis; kidney carcinoma; benign bladder tumor ^3^	CYFRA21-1
Grossman et al. 2005 [48]	USA	2001-2002	Prospective	Multi-centre	Trial	H&Co	79	1252	685	567	(18–96)	58.7 ± 14.3	759M:572F	-	120 Black non-Hispanic; 1089 White non-Hispanic; 84 Hispanic; 26 Asian;6 Other;6 Unknown	x	-	x	Benign prostatic hypertrophy; prostatitis; cystitis; inflammation; trigonitis; urinary tract infection; erythema; hyperplasia; squamous; metaplasia; cysts and polyps; calculi; trabeculations; other benign disease; kidney and genitourinary; other cancer history; non bladder; other active cancer; non bladder	NMP-22
Horstmann et al. 2012 [49]	Germany	NA	Prospective	Single centre	Cohort	U	377	1177	-	-	(23–96)	67	1169M:385F	-	605 with previous bladder cancer diagnosis^4^	x		-	x	UroVysion
																				uCyt+
																				NMP-22
Karnes et al. 2012 [50]	NA	2009–2011	Prospective	Multi-centre	Observational	H&C	58	690	-	-	-	C (71 ± 10.0) Con (64 ± 9.6)	450M:298F	444Y:300N: 4 Unknown	-	x		-	-	MADR Assay
																				FGFR3
																				FGFR3 + Cytology
Kelly et al. 2012 [51]	UK	NA	Prospective	Multi-centre	Observational	H	222	1455	755	700	(49–73)	60.7 ± 16.3 median:63	1040M:637F	-	-	x	x	x	Urolithiasis; other malignancies; other benign conditions ^3^	Mcm5
																				NMP-22
																				Mcm5 + NMP-22
Liu et al. 2016 [52]	China	2010–2014	Prospective	Multi-centre	Cohort	H	141	135	-	135	-	-	-	-	-	-	-	-	-	Gamma-synuclein (SNCG)
Meiers et al. 2007 [53]	Belgium	NA	NA	NA	NA	U	170	454	454	0	-	-	-	-	-	-	-	-	-	UroVysion
O’Sullivan et al. 2012 [54]	Australasia (NZ/Australia)	NA	Prospective	Multi-centre	Cohort	H	66	419	255	164	(59–77)	median:69	389M:96F	76 Current: 215 Former: 194 Never	423 European; 33 Māori; 29 Other	x	x	-	-	uRNA-D
																				NMP-22
																				NMP-22 BladderChek
																				Cxbladder
Oertl et al. 2007 [55]	NA	NA	Prospective	NA	NA	U	56	51	51	0	-	-	-	-	Benign bladder tumour	-	-	-	Cystitis; pyelonephritis; urethritis	NMP-22
Piaton et al. 2003 [56]	France	NA	Prospective	Multi-centre	NA	H	59	156	156	0	(32–92)	66.2 ± 12.8	550M:144F ^5^	-	-	x	x	x	Dysuria; pollakiuria; cystitis; cystalgia; other condition	uCyt+
																				uCyt+ (+) Cytology
Poulakis et al. 2001 [57]	NA	NA	Prospective	NA	NA	U	406	333	212	121	(67–90)	66.7	485M:254F	-	386 with previous bladder carcinoma ^4^	x		-	-	NMP-22
																				BTAstat
Saad et al. 2002 [58]	UK	NA	Prospective	Single centre	NA	H	52	68	53	15	(30–88)	69.5 ± 11.1	100M:20F	-	-	-	-	x	Benign prostatic hyperplasia; urethral strictures; renal stone; cystodistension; 12 prostate cancer; 2 renal cell carcinoma	BTAstat
																				NMP-22
Sajid et al. 2020 [59]	Pakistan	2018–2019	NA	Single centre	Cross-sectional	H	215	165	-	-	(24–75)	53.08 ± 12.41	300M:80F	-	-	-	-	-	-	NMP-22
																				NMP-22 + Cytology
Sanchez-Carbayo et al. 2000 [60]	Spain	NA	Prospective	Single centre	NA	H	52	137	45	32	-	-	311M:90F ^5^	-	-	-	x	x	Benign diseases; lithiasis; Stenosis; benign prostate hyperplasia	BTF
																				BTF normalised
																				CK18
																				CK18 normalised
Sarosdy et al. 2006 [61]	NA	NA	Prospective	Multi-centre	Trial	U	51	422	-	-	(40–97)	63	298M:199F ^6^	265 Current or former: 207 Never: 1 Unknown	440 White; 26 Black; 15 Hispanic; 4 Asian; 12 Other race ^6^	x		-	Inflammation or infection; benign prostatic hyperplasia; stones; urethral stricture; bladder polyp	UroVysion
Shang et al. 2021 [62]	China	2020–2021	Prospective	Single centre	NA	H	128	136	136	0	C (32–89) Con (35–84)	C (67.5 ± 10.2) Con (68.9 ± 13)	190M:74F	-	-	x		-	Irritative bladder symptoms, abdominal pain, hydronephrosis on the affected side	UroVysion
Todenhöfer et al. 2013 [63]	Germany	NA	NA	Single centre	Cohort	U	115	693	-	-	(21–92)	median:67	645M:163F	-	-	x		x	Irritative voiding symptoms	UroVysion
																				uCyt+
																				NMP-22
																				UroVysion + uCyt+
																				UroVysion + NMP-22
																				uCyt+ (+) NMP-22
																				UroVysion + uCyt+ (+) NMP-22
																				UroVysion + Cytology
																				uCyt+ (+) Cytology
																				NMP-22 + Cytology
																				uCyt+ (+) UroVysion + Cytology
																				UroVysion + NMP-22+ Cytology
																				uCyt+ (+) NMP-22 + Cytology
																				UroVysion + uCyt+ (+) NMP22+ Cytology
																				Base model (age and grade of haematuria) + UroVysion
																				Base model (age and grade of haematuria) + UroVysion + Cytology
																				Base model (age and grade of haematuria) + UroVysion + uCyt+
																				Base model (age and grade of haematuria) + UroVysion + uCyt+ (+) Cytology)
Todenhöfer et al. 2013 [64]	Germany	2004–2009	NA	Single centre	Cohort	U	502	1611	1611	0	(23–96)	median:66	1599M:514F	-	-	-	-	x	-	UroVysion
																				uCyt+
																				NMP-22
Todenhöfer et al. 2013 [65]	Germany	2004–2009	Retrospective	Single centre	Cohort	U	543	1822	-	-	(18–97)	median:65	1776M:589F	-	-	x		-	Irritative voiding symptoms	UroVysion
																				uCyt+
																				NMP-22
van Kessel et al. 2016 [66]	The Netherlands	2006–2013	Prospective	Single centre	Cohort	H	74	80	80	0	C (38–91) Con (21–86)	C 68 Con 58	109M:45F	-	-	x		-	-	FGFR3
																				TERT
																				FGFR3, TERT, HRAS
																				TWIST1
																				OTX1 probe 1
																				OTX1 probe 2
																				OTX1 probe 3
																				OTX1 probe 4
																				ONECUT 2 probe 1
																				ONECUT 2 probe 2
																				ONECUT 2 probe 3
																				ONECUT 2 probe 4
																				ONECUT 2 probe 5
																				OTX1 all combined
																				ONECUT2 all combined
																				ONECUT probes 1 + 4
																				Optimal (age, FGFR3, TERT, HRAS, ONECUT2 probes 1 + 4, OTX1 probe 2, Twist)
van Kessel et al. 2017 [67]	Sweden, Spain, The Netherlands	NA	Prospective	Multi-centre	Cohort	H	97	103	103	0	C (38–110) Con (50–82)	C 71 Con 62	181M:19F	-	-	x		-	-	FGFR3, TERT, HRAS and OTX1, ONECUT2, TWIST1
van Kessel et al. 2020 [68]	The Netherlands	2015–2017	Prospective	Multi-centre	Cohort	H	112	726	-	-	(19–96)	median:66	485M:353F	221 Current: 185 Former: 258 Never: 174 Not reported	-	x	x	-	-	FGFR3, TERT, Harvey HRAS and OTX1, ONECUT2, TWIST1
																				Extended Model consisting of: Existing model (univariate analysis incl. age, mutation, methylation) + type of haematuria + gender
																				Optimal model consisting of: Existing model (univariate analysis incl. age, mutation, methylation) + type of haematuria
van Valenberg et al. 2021 [2]	USA	2015	Prospective	Multi-centre	Cohort	U	59	769	-	-	(23–84)	65 ± 13	467M:361F	139 Current: 288 Former: 401 Never	698 Caucasian; 80 Black; 32 Hispanic; 18 Other	x	x	-	-	Xpert
																				UroVysion
Virk et al. 2017 [69]	USA	2008–2014	Retrospective	Single centre	Cohort	H	181	196	196	0	(24–93)	mean & median: 67	259M:118F	-	-	x		-	Other Symptoms; miscellaneous indications	UroVysion
Ward et al. 2022 [70]	UK	NA	Prospective	Multi-centre	NA	H	68	147	-	-	-	median: C 72.5: Con 60	137M:78F	-	-	-	-	-	Calculi, benign prostatic hyperplasia, cystitis, inflammation, prostate cancer, and kidney cancer	TERT (promoter), FGFR3, PIK3CA, TP53, ERCC2, RHOB, ERBB2, HRAS, RXRA, ELF3, CDKN1A, KRAS, KDM6A, AKT1, FBXW7, ERBB3, SF3B1, CTNNB1, BRAF, C3orf70, CREBBP, CDKN2A, and NRAS
Wu et al. 2020 [71]	China	2015–2016	Prospective	Single centre	Case-control	H	53	58	58	0	C (61.5–74): Con (48–63.3)	median: C 68: Con 55.5	83M:28F	-	-	x		x	Urinary calculi; benign prostatic hyperplasia	HOXA9
																				PCDH17
																				POU4F2
																				ONECUT2
																				HOXA9 + PCDH17 + POU4F2 + ONECUT2
Zhou et al. 2019 [72]	China	2007–2008	Prospective	Multi-centre	Trial	H	3640	319			(15–97)	63.04 ± 13.31	3021M:938F	-	-	x	-	-	Inflammation; Renal tumor; benign bladder tumor; bladder tumor with non-transitional cell origin ^3^	UroVysion

C cases, HC healthy controls, Con controls, NM non-malignant, NA not available, Y yes, N no, VH visible haematuria, NVH non-visible haematuria, UTIs urinary tract infections. ^1^ mean (SD) reported where applicable, unless otherwise stated, ^2^ numbers reported only for cases, ^3^ malignancies/tumors identified following examination, ^4^ number of participants with prior cancer diagnosis (less than half of included sample), ^5^ number of all participants recruited (cases and controls reported only for the group of interest), ^6^ number of all participants enrolled (prior exclusion), ^7^ also referred to as ImmunoCyt+.

**Table 2 cancers-15-00709-t002:** List of Biomarkers/tests identified in the included studies.

No	Biomarker/Test	Biomarker Category			Description	Study	Test Platform	Report		Sample Source		Threshold Used (Where Available/Appropriate)	Performance Measures Reported	Included in Meta-Analysis
		*Proteins*	*mRNA*	*DNA*				*Individual Markers*	*Multiple Markers*	*Urine*	*Serum*			
**INDIVIDUAL MARKERS**														
1	ADXBLADDER/Mcm5	x			Protein expression of Minichromosome Maintenance protein 5	[37]	ELISA ^1^	x		x			SN, SP, NPV, AUC, Accuracy, FP, FN	No
						[51]	Immunofluorometric Assay	x		x		2150-cell;8500-cell	SN, SP, PPV, NPV	No
2	ANG	x			Protein expression of Angiogenin	[41]	ELISA	x		x		425 pg/mg	SN, SP, PPV, NPV, Accuracy	No
3	BCL2			x	Methylation of BCL2 gene	[34]	qPCR ^2^	x		x			SN, SP, PPV, NPV	No
4	bFGF	x			Protein expression of Basic Fibroblast Growth Factor	[41]	ELISA	x				19,444 fmol/mg	SN, SP, PPV, NPV, Accuracy	No
5	BTAstat	x			Protein expression of Complement factor H-related protein	[32]	Lateral Flow Test	x		x			SN, SP, FP	Yes
						[57]	Lateral Flow Test	x		x			SN, SP, PPV, NPV, Accuracy	
						[58]	Lateral Flow Test	x		x			SN, SP, PPV, Accuracy, FP, FN	
6	BTF	x			Protein expression of Bladder Tumour Fibronectin	[60]	Chemiluminescence Immunoassay	x		x		52.8 µg/L	SN, SP, PPV, NPV, Accuracy	No
												77 µg/g (when normalising with urine CREA)		
7	CCNA1			x	Methylation of CCNA1 gene	[34]	qPCR	x		x			SN, SP, PPV, NPV	No
8	CK18	x			Protein expression of Cytokeratin 18	[60]	Chemiluminescence Immunoassay	x		x		7.4 µg/L	SN, SP, PPV, NPV, Accuracy	No
												7 µg/g (when normalising with urine CREA)		
9	CK20		x		mRNA expression of Cytokeratin 20	[43]	RT-PCR ^3^	x		x			SN, SP, PPV, NPV, Accuracy	No
10	CYFRA21-1	x			Protein expression of Cytokeratin 19 fragments	[47]	Chemiluminescence Immunoassay	x		x		2.28 ng/mL	SN, SP, AUC	No
											x	62.74 ng/mL	SN, SP, AUC	
										x	x	0.66 ng/mL	SN, SP, AUC	
11	EMA	x			Protein expression of Epithelial Membrane Antigen	[30]	ELISA	x		x		1.8 Ug/mL	SN, SP, PPV, NPV, AUC, Efficiency	No
12	EOMES			x	Methylation of EOMES gene	[34]	qPCR	x		x			SN, SP, PPV, NPV	No
13	FGFR3			x	Mutation of FGFR3 gene		ddPCR ^4^	x		x		3	SN, SP, PPV, NPV	
						[50]	qPCR	x		x		“negative”	SN, SP, NPV	
						[66]	SNaPshot assay	x		x			SN, SP, Accuracy, FP, FN	
14	FN	x			Protein expression of Fibronectin	[43]	ELISA	x		x		41.7 ng/mg	SN, SP, PPV, NPV, Accuracy	No
15	HGF	x			Protein expression of Hepatocyte Growth Factor	[41]	ELISA	x		x		1820 pg/mg	SN, SP, PPV, NPV, Accuracy	No
16	HOXA9			x	Methylation of HOXA9 gene	[71]	High Resolution Melting	x		x			SN, SP, PPV, NPV	No
17	HURP		x		mRNA expression of Hepatoma Up-regulated Protein	[45]	RT-PCR	x		x		0.0132	SN, SP, PPV, NPV	No
						[46]	RT-PCR & AuNP ^5^ assay	x		x			SN, SP, PPV, NPV, Accuracy, FP, FN	
18	HYAL-1	x			Hyaluronidase 1 activity	[44]	Zymography	x		x		13.8 µU/mg	SN, SP, PPV, NPV, AUC, Accuracy	No
			x		mRNA expression of Hyaluronidase 1 (Qual/Semi-quant)	[42]	RT-PCR	x		x		0.25 (Qual)	SN, SP, PPV, NPV, Accuracy	
19	MMP-2	x			Protein expression of Matrix metalloproteinase 2	[39]	ELISA	x		x		1.9 ng/mg	SN, SP	No
20	MMP-9	x			Protein expression of Matrix metalloproteinase 9		ELISA	x		x		8.7 ng/mg	SN, SP	No
21	NMP-22	x			Protein expression of Nuclear Matrix Protein 22	[48]	Lateral Flow Test (BladderChek)	x		x		10 IU/mL	SN, SP, PPV, NPV	Yes
						[54]	Lateral Flow Test (BladderChek)	x		x			SN, SP	
						[59]	Lateral Flow Test (BladderChek)	x		x			SN, SP, PPV, NPV, AUC, Accuracy, FP, FN	
						[32]	ELISA	x		x		3.6 IU/mL	SN, SP, FP	
						[49]	ELISA	x		x		10 IU/mL	SN, SP, Accuracy, FP	
						[51]	ELISA	x		x		10 IU/mL	SN, SP, PPV, NPV, AUC, Accuracy	
						[54]	ELISA	x		x		7.5 IU/mL	SN, SP, AUC	
						[55]	ELISA	x		x		7.5 IU/mL	SN	
						[57]	ELISA	x		x		8.25 IU/mL	SN, SP, PPV, NPV, Accuracy	
						[63]	ELISA	x		x		≥10 IU/mL	SN, SP, PPV, NPV, AUC	
						[64]	ELISA	x		x		≥10 IU/mL	SN, SP	
						[65]	ELISA	x		x		≥10 IU/mL	SN, SP	
						[58]	Unclear	x		x		≥10 IU/mL	SN, SP, PPV, Accuracy, FP, FN	
22	NMP-52	x			Protein expression of Nuclear Matrix Protein 52	[30]	ELISA	x		x		2.8 Ug/mL	SN, SP, PPV, NPV, AUC, Efficiency	No
23	OTX1			x	Methylation of OTX1 gene (probe 1/probe 2/probe 3/probe 4/all combined)	[66]	SNaPshot assay	x		x			SN, SP, AUC	No
24	ONECUT2			x	Methylation of ONECUT2 gene	[34]	qPCR	x		x			SN, SP, PPV, NPV	No
						[71]	High Resolution Melting	x		x			SN, SP, PPV, NPV	No
					Methylation of ONECUT2 gene (probe 1/probe 2/probe 3/probe 4/probe 5/probe 1 + 4/all combined)	[66]	SNaPshot assay	x		x			SN, SP, AUC	No
25	PCDH17			x	Methylation of PCDH17 gene	[71]	High Resolution Melting	x		x			SN, SP, PPV, NPV	No
26	POU4F2			x	Methylation of POU4F2 gene		High Resolution Melting	x		x			SN, SP, PPV, NPV	No
27	RAB-B2			x	Methylation of RAB-B2 gene	[44]	PCR	x		x		0.065	SN, SP, PPV, NPV, AUC, Accuracy	No
28	SALL3			x	Methylation of SALL3 gene	[34]	qPCR	x		x		0.5	SN, SP, PPV, NPV	No
29	SNCG	x			Protein expression of Gamma-Synuclein	[52]	ELISA	x				1.874 ng/mL	SN, SP, PPV, NPV, AUC, Accuracy	No
30	SPARC	x			Protein expression of Secreted Protein Acidic and Rich in Cysteine	[33]	ELISA	x		x		>0 ng/mL^−1^	SN, SP, PPV, NPV, AUC	No
31	Survivin		x		mRNA expression of Survivin (Qual/Semi-quant)	[42]	RT-PCR	x		x		0.25 (Qual)	SN, SP, PPV, NPV, Accuracy	No
32	Telomerase (TERT; hRT; hTERT; RTA)			x	Mutation of TERT gene	[34]	ddPCR	x		x		5 (TERT C250T)/3 (TERT C228T)	SN, SP, PPV, NPV	No
						[66]	SNaPshot assay	x		x			SN, SP, AUC	
			x		mRNA expression of Telomerase (hRT/hTERT)	[38]	RT-PCR	x		x			SN, SP, PPV, NPV, Accuracy, FP, FN	No
		x			Telomerase activity (RTA)		qPCR	x		x		18.08 copies per 2µl	SN, SP, PPV, NPV, AUC, Accuracy, FP, FN	
							TRAP ^6^ assay	x		x		0.046	SN, SP, PPV, NPV, AUC, Accuracy, FP, FN	
					Telomerase activity	[43]	TRAP assay	x					SN, SP, PPV, NPV, Accuracy	No
						[32]	TRAP assay	x		x			SN, SP, FP	
33	TGF-β1	x			Protein expression of Transforming Growth Factor Beta 1	[40]	ELISA	x				25 pg/mg	SN, SP, PPV, NPV, AUC, Accuracy	No
34	TIMP-2	x			Protein expression of Tissue Inhibitor of Metalloproteinase 2	[39]	ELISA	x		x		4.49 ng/mg	SN, SP	No
35	TWIST1			x	Methylation of TWIST1 gene	[66]	qPCR	x		x			SN, SP, AUC	No
36	VEGF	x			Protein expression of Vascular Endothelial Growth Factor	[40]	ELISA	x		x		38 pg/mg	SN, SP, PPV, NPV, AUC, Accuracy	No
						[41]	ELISA	x		x		52ng/mg	SN, SP, AUC, FP, FN	No
37	VIM			x	Methylation of VIM gene	[34]	qPCR	x		x		2	SN, SP, PPV, NPV	No
**MULTIPLE MARKERS**														
1	CCNA1 + ONECUT2 + BCL2 + EOMES + SALL3 + VIM			x	Methylation of CCNA1, ONECUT2, BCL2, EOMES, SALL3 and VIM genes	[34]	qPCR		x	x			SN, SP	No
2	CK20 + RTA	x	x		mRNA expression of Cytokeratin 20 + telomerase activity	[43]	RT-PCR + TRAP assay		x	x			SN, SP, PPV, NPV, Accuracy	No
3	CK20 + FN + RTA	x	x		mRNA expression of Cytokeratin 20 + protein expression of Fibronectin + telomerase activity		RT-PCR + ELISA + TRAP assay		x	x			SN, SP, PPV, NPV, Accuracy	No
4	CxBladder		x		mRNA expression of CDK1, HOXA13, MDK, IGFBP5 and CXCR2 genes	[35]	qPCR		x	x		<4	SN, SP, NPV, Accuracy	No
						[54]						predefined as the classifier scores which gave 85% to 90% SP in this cohort	SN, SP, AUC	
5	EMA + NMP-52	x			Protein expression of Epithelial Membrane Antigen and Nuclear matrix protein 52	[30]	ELISA		x	x		4.0 Ug/mL	SN, SP, PPV, NPV, AUC, Efficiency	No
6	FGFR3, TERT, HRAS			x	Mutation of FGFR3, TERT and HRAS genes	[66]	SNaPshot assay		x	x			SN, SP, AUC	No
7	FGFR3, TERT, HRAS and OTX1, ONECUT2, TWIST1			x	Mutation of FGFR3, TERT and HRAS genes and methylation of OTX1, ONECUT2 and TWIST1 genes	[67]	SNaPshot assay and qPCR		x	x		0.1917196	SN, SP	No
						[68]	SNaPshot assay and qPCR		x	x			SN, SP, PPV, NPV, Accuracy	No
8	HOXA9 + PCDH17 + POU4F2 + ONECUT2			x	Methylation of HOXA9, PCDH17, POU4F2 and ONECUT2 gene	[71]	High Resolution Melting		x				SN, SP, PPV, NPV, AUC	No
9	HYAL-1 + Survivin		x		mRNA expression of Hyaluronidase 1 + Survivin (Qual/Semi-quant)	[42]	RT-PCR		x	x			SN, SP, PPV, NPV, Accuracy	No
10	MADR Assay	x		x	Methylation of TWIST1 and NID2 genes, mutation of FGFR3 gene and protein expression of matrix metalloproteinase 2	[50]	PCR + qPCR + ELISA		x	x		TWIST1 < 139K; NID2 < 680; MMP-2 < 1.100 ng/mL; FGF3 negative	SN, SP, NPV	No
11	Mcm5 + NMP-22	x			Protein expression of Minichromosome Maintenance protein 5 and Nuclear Matrix Protein 22	[51]	Immunofluorometric Assay + ELISA		x	x		4200-cell; 2800-cell; 1900-cell; 1000-cell + 10 IU/mL	SN, SP	No
12	MMP-2 + TIMP-2	x			Protein expression of Matrix metalloproteinase 2 + Tissue inhibitor of metalloproteinase 2	[39]	ELISA		x	x		0.93 ng/mg	SN, SP	No
13	MMP-9 + TIMP-2	x			Protein expression of Matrix metalloproteinase 9 + Tissue inhibitor of metalloproteinase 2	[39]	ELISA		x	x		3.81 ng/mg	SN, SP	No
14	MMP-2/MMP-9	x			MMP-2/MMP-9 activity	[39]	Zymography		x	x			SN, SP	No
15	RAB-B2 + HYAL-1	x		x	Methylation of RAB-B2 + Hyaluronidase 1 activity	[44]	PCR + Zymography		x	x			SN, SP, PPV, NPV, Accuracy	No
16	Telomerase (RTA + hRT; RTA + hTERT; hTR + hTERT; RTA + hRT + hTERT)	x	x		Telomerase activity (RTA) + telomerase mRNA expression (hRT)	[38]	TRAP assay + RT-PCR		x	x			SN, SP, PPV, NPV, Accuracy	No
					Telomerase activity (RTA) + telomerase mRNA expression (hTERT)		TRAP assay + qPCR		x	x			SN, SP, PPV, NPV, Accuracy	No
			x		Telomerase mRNA expression (hTR + hTERT)		RT-PCR + qPCR		x	x			SN, SP, PPV, NPV, Accuracy	No
		x	x		Telomerase activity (RTA) + telomerase mRNA expression (hTR + hTERT)		TRAP assay + RT-PCR + qPCR		x	x			SN, SP, PPV, NPV, Accuracy	No
17	TERT (promoter), FGFR3, PIK3CA, TP53, ERCC2, RHOB, ERBB2, HRAS, RXRA, ELF3, CDKN1A, KRAS, KDM6A, AKT1, FBXW7, ERBB3, SF3B1, CTNNB1, BRAF, C3orf70, CREBBP, CDKN2A, and NRAS			x	Mutation of TERT (promoter), FGFR3, PIK3CA, TP53, ERCC2, RHOB, ERBB2, HRAS, RXRA, ELF3, CDKN1A, KRAS, KDM6A, AKT1, FBXW7, ERBB3, SF3B1, CTNNB1, BRAF, C3orf70, CREBBP, CDKN2A, and NRAS	[70]	NGS ^7^		x	x		a positive test was defined as detection of any one of the 443 mutations in a cpDNA sample at >0.9% VAF for chr5:129528A/G or >0.5% VAF for all other coordinates.	SN, SP	No
18	TERT, FGFR3, SALL3, ONECUT2, CCNA1, BCL2, EOMES, and VIM			x	Mutation of TERT and FGFR3 genes plus methylation of SALL3, ONECUT2, CCNA1, BCL2, EOMES, and VIM genes	[34]	ddPCR/qPCR		x	x			SN, SP, PPV, NPV, AUC	No
19	TERT + ONECUT2			x	Mutation of TERT gene plus methylation of ONECUT2 genes		ddPCR/qPCR		x	x			SN, SP	No
20	TERT + FGFR3			x	Mutation of TERT and FGFR3 genes		ddPCR/qPCR		x	x			SN, SP	No
21	TERT + FGFR3 + CCNA1			x	Mutation of TERT and FGFR3 genes plus methylation of CCNA1 genes		ddPCR/qPCR		x	x			SN, SP	No
22	TERT + FGFR3 + CCNA1 + ONECUT2			x	Mutation of TERT and FGFR3 genes plus methylation of CCNA1 and ONECUT2 genes		ddPCR/qPCR		x	x			SN, SP	No
23	TERT + FGFR3 + CCNA1+ ONECUT2 + BCL2			x	Mutation of TERT and FGFR3 genes plus methylation of CCNA1, ONECUT2 and BCL2 genes		ddPCR/qPCR		x	x			SN, SP	No
24	TERT + FGFR3 + CCNA1 + ONECUT2 + BCL2 + EOMES			x	Mutation of TERT and FGFR3 genes plus methylation of CCNA1, ONECUT2, BCL2 and EOMES genes		ddPCR/qPCR		x	x			SN, SP	No
25	TERT + FGFR3 + CCNA1 + ONECUT2 + BCL2 + EOMES + SALL3			x	Mutation of TERT and FGFR3 genes plus methylation of CCNA1, ONECUT2, BCL2, EOMES and SALL3 genes		ddPCR/qPCR		x	x			SN, SP	No
26	TGF-β1 + VEGF	x			Protein expression of Transforming growth factor beta 1 + Vascular endothelial growth factor	[40]	ELISA		x	x			SN, SP, PPV, NPV, Accuracy	No
27	uCyt+^9^	x			Protein expression of Carcinoembryonic antigen and sulphate mucin glycoproteins	[36]	Immunofluorescence		x	x			SN, SP, FP, FN	Yes
						[49]	Immunofluorescence		x	x		at least 1 clear positive cell	SN, SP, Accuracy, FP	
						[56]	Immunofluorescence		x	x			SN, SP, FP	
						[63]	Immunofluorescence		x	x		at least 1 clear positive cell	SN, SP, PPV, NPV, AUC	
						[64]	Immunofluorescence		x	x		at least 1 clear positive cell	SN, SP	
						[65]	Immunofluorescence		x	x		at least 1 clear positive cell	SN, SP, Accuracy, FP, FN	
28	uCyt+ (+) NMP-22	x			Protein expression of Carcinoembryonic antigen and Sulphate mucin glycoproteins and nuclear matrix protein 22	[63]	Immunofluorescence + ELISA		x	x		uCyt+ => 1 clear positive cell/NMP22 = > 10 IU/mL	SN, SP, PPV, NPV, AUC	No
29	uRNA-D		x		mRNA expression of CDC2, HOXA13, MDK and IGFBP5 genes	[54]	qPCR		x	x		predefined as the classifier scores which gave 85% SP in this cohort	SN, SP, AUC	No
30	UroVysion			x	Aneuploidy of Chromosomes 3, 7 and 17 and loss of chromosome locus 9p21	[49]	FISH ^8^		x	x			SN, SP, Accuracy, FP	Yes
						[53]	FISH		x	x			SN, SP	
						[61]	FISH		x	x			SN, SP, PPV, NPV, Accuracy	
						[62]	FISH		x	x			SN, SP	
						[63]	FISH		x	x			SN, SP, PPV, NPV, AUC	
						[64]	FISH		x	x			SN, SP	
						[65]	FISH		x	x			SN. SP,	
						[2]	FISH		x	x			SN, SP, PPV, NPV, Accuracy	
						[69]	FISH		x	x			SN, SP, PPV, NPV, Accuracy	
						[72]	FISH		x	x			SN, SP, PPV, NPV, AUC, FP, FN	
31	UroVysion + uCyt+	x		x	Aneuploidy of Chromosomes 3, 7 and 17 and loss of chromosome locus 9p21 + protein expression of Carcinoembryonic antigen and Sulphate mucin glycoproteins	[63]	FISH + Immunofluorescence		x	x			SN, SP, PPV, NPV, AUC	No
32	UroVysion + NMP-22	x		x	Aneuploidy of Chromosomes 3, 7 and 17 and loss of chromosome locus 9p21 + protein expression of nuclear matrix protein 22	[63]	FISH + ELISA		x	x		NMP-22 => ≥10IU/mL	SN, SP, PPV, NPV, AUC	No
33	UroVysion + uCyt+ (+) NMP-22	x		x	Aneuploidy of Chromosomes 3, 7 and 17 and loss of chromosome locus 9p21 + protein expression of Carcinoembryonic antigen and Sulphate mucin glycoproteins and nuclear matrix protein 22	[63]	FISH + Immunofluorescence + ELISA		x	x			SN, SP, PPV, NPV, AUC	No
34	Xpert		x		mRNA expression of ABL1, CRH, IGF2, UPK1B and ANXA10	[2]	qPCR		x	x			SN, SP, PPV, NPV, AUC, Accuracy	No
**BIOMARKER/TESTS and CYTOLOGY**														
1	CK20 + Cytology		x		mRNA expression of Cytokeratin 20 + Cytology	[43]	RT-PCR		x	x			SN, SP, PPV, NPV, Accuracy	No
2	CK20 + RTA + Cytology	x	x		mRNA expression of Cytokeratin 20 + telomerase activity + Cytology		RT-PCR + TRAP assay		x	x			SN, SP, PPV, NPV, Accuracy	No
3	FGFR3 + Cytology			x	Mutation of FGFR3 gene + Cytology	[50]	qPCR		x	x			SN, SP, PPV	No
4	FN + Cytology	x			Protein expression of Fibronectin + Cytology	[43]	ELISA		x	x			SN, SP, PPV, NPV, Accuracy	No
5	FN + CK20 + Cytology	x	x		Protein expression of Fibronectin + mRNA expression of Cytokeratin 20 + Cytology		ELISA + RT-PCR		x	x			SN, SP, PPV, NPV, Accuracy	No
6	FN + CK20 + RTA + Cytology	x	x		Protein expression of Fibronectin + mRNA expression of Cytokeratin 20 + telomerase activity + Cytology		ELISA + RT-PCR + TRAP assay		x	x			SN, SP, PPV, NPV, Accuracy	No
7	HURP + Cytology		x		mRNA expression of Hepatoma Up-regulated Protein + Cytology	[45]	RT-PCR		x	x		0.0132	SN, SP, PPV, NPV, Accuracy	No
						[46]	RT-PCR & AuNP assay		x	x			SN, SP, PPV, NPV, Accuracy	No
9	HYAL-1 + Cytology		x		mRNA expression of Hyaluronidase 1 + Cytology (Qual/Semi-quant)	[42]	RT-PCR		x	x			SN, SP, PPV, NPV, Accuracy	No
		x			Hyaluronidase activity + Cytology	[44]	Zymography		x	x			SN, SP, PPV, NPV, Accuracy	No
10	HYAL-1 + Survivin + Cytology		x		mRNA expression of Hyaluronidase 1 and Survivin + Cytology (Qual/Semi-quant)	[42]	RT-PCR		x	x			SN, SP, PPV, NPV, Accuracy	No
11	MMP-2 and MMP-9 zymography + Cytology	x			Activity of Matrix metalloproteinase 2 and Matrix metalloproteinase 9 + Cytology	[39]	Zymography		x	x			SN, SP	No
12	MMP-2 + Cytology	x			Protein expression of Matrix metalloproteinase 2 + Cytology		ELISA		x	x		1.9 ng/mg	SN, SP	No
13	MMP-9 + Cytology	x			Protein expression of Matrix metalloproteinase 9 + Cytology		ELISA		x	x		8.7 ng/mg	SN, SP	No
14	MMP-2 + TIMP-2 ratio + Cytology	x			Protein expression of Matrix metalloproteinase 2 and Tissue inhibitor of metalloproteinase 2 + Cytology		ELISA		x	x		0.93 ng/mg	SN, SP	No
15	MMP-9 + TIMP-2 ratio + Cytology	x			Protein expression of Matrix metalloproteinase 9 + Tissue inhibitor of metalloproteinase 2 + Cytology		ELISA		x	x		3.81 ng/mg	SN, SP	No
16	NMP-22 + Cytology	x			Protein expression of Nuclear Matrix Protein 22 + Cytology	[59]	Lateral Flow Test (BladderChek)		x	x			SN, SP, PPV, NPV, AUC, Accuracy, FP, FN	No
						[63]	ELISA		x	x		≥10 IU/mL	SN, SP, PPV, NPV, AUC	No
17	RAB-B2 + Cytology			x	Methylation of RAB-B2 gene + Cytology	[44]	PCR		x	x		0.065	SN, SP, PPV, NPV, Accuracy	No
18	RAB-B2 + HYAL-1 + Cytology	x		x	Methylation of RAB-B2 gene + Hyaluronidase activity + Cytology		PCR + Zymography		x	x			SN, SP, PPV, NPV, Accuracy	No
19	Survivin + Cytology		x		mRNA expression of Survivin + Cytology	[42]	RT- PCR		x	x			SN, SP, PPV, NPV, Accuracy	No
20	**Telomerase + Cytology** (RTA + Cytology; hRT + Cytology; hTERT + Cytology; RTA + hRT + Cytology; RTA + hTERT + Cytology; hTR + hTERT + Cytology; RTA + hRT + HTERT + Cytology)	x			Telomerase activity (RTA) + Cytology	[38]	TRAP assay		x	x			SN, SP, PPV, NPV, Accuracy	No
						[43]	TRAP assay		x				SN, SP, PPV, NPV, Accuracy	No
			x		mRNA expression of Telomerase (hRT/hTERT) + Cytology	[38]	RT-PCR		x	x			SN, SP, PPV, NPV, Accuracy	No
							qPCR		x	x			SN, SP, PPV, NPV, Accuracy	No
		x	x		Telomerase activity (RTA) + mRNA expression of Telomerase (hRT/hTERT) + Cytology		TRAP assay + RT-PCR		x	x			SN, SP, PPV, NPV, Accuracy	No
							TRAP assay + qPCR		x	x			SN, SP, PPV, NPV, Accuracy	No
			x		mRNA expression of Telomerase (hTR + hTERT) + Cytology		RT-PCR + qPCR		x	x			SN, SP, PPV, NPV, Accuracy	No
		x	x		Telomerase activity (RTA) + mRNA expression of Telomerase (hTR + hTERT) + Cytology		TRAP assay + RT-PCR + qPCR		x	x			SN, SP, PPV, NPV, Accuracy	No
21	TGF-β1 + Cytology	x			Protein expression of Transforming Growth Factor Beta 1 + Cytology	[40]	ELISA		x	x			SN, SP, PPV, NPV, Accuracy	No
22	TGF-β1 + VEGF + Cytology	x			Protein expression of Transforming Growth Factor Beta 1 + Vascular Endothelial Growth Factor + Cytology		ELISA		x	x			SN, SP, PPV, NPV, Accuracy	No
23	TIMP-2 + Cytology	x			Protein expression of Tissue inhibitor of metalloproteinase 2 + Cytology	[39]	ELISA		x	x		4.49 ng/mg	SN, SP	No
24	uCyt+ + Cytology	x			Protein expression of Carcinoembryonic Antigen and Sulphate Mucin Glycoproteins + Cytology	[56]	Immunofluorescence		x	x			SN	No
						[63]	Immunofluorescence		x	x			SN, SP, PPV, NPV, AUC	No
25	uCyt+ (+) NMP-22 + Cytology	x			Protein expression of Carcinoembryonic antigen, Sulphate mucin glycoproteins and Nuclear matrix protein 22 + Cytology		Immunofluorescence + ELISA		x	x			SN, SP, PPV, NPV, AUC	No
26	UroVysion + Cytology			x	Aneuploidy of Chromosomes 3, 7 and 17 and loss of chromosome locus 9p21		FISH		x	x			SN, SP, PPV, NPV, AUC	No
27	UroVysion + uCyt+ (+) Cytology	x		x	Aneuploidy of Chromosomes 3, 7 and 17 and loss of chromosome locus 9p21 + protein expression of Carcinoembryonic antigen and sulphate mucin glycoproteins + Cytology		FISH + Immunofluorescence		x	x			SN, SP, PPV, NPV, AUC	No
28	UroVysion + NMP-22 + Cytology	x		x	Aneuploidy of Chromosomes 3, 7 and 17 and loss of chromosome locus 9p21 + protein expression of nuclear matrix protein 22 + Cytology		FISH + ELISA		x	x			SN, SP, PPV, NPV, AUC	No
29	UroVysion + uCyt+ (+) NMP-22 + Cytology	x		x	Aneuploidy of Chromosomes 3, 7 and 17 and loss of chromosome locus 9p21 + protein expression of Carcinoembryonic antigen, Sulphate mucin glycoproteins and nuclear matrix protein 22 + Cytology		FISH + Immunofluorescence + ELISA		x	x			SN, SP, PPV, NPV, AUC	No
30	VEGF + Cytology	x			protein expression of Vascular endothelial growth factor + Cytology	[40]	ELISA		x	x			SN, SP, PPV, NPV, Accuracy	No
**BIOMARKER/TESTS and PREDICTION MODELS**														
1	Base model (age and grade of haematuria) + UroVysion			x	Aneuploidy of Chromosomes 3, 7 and 17 and loss of chromosome locus 9p21	[63]	FISH		x				AUC	No
2	Base model (age and grade of haematuria) + UroVysion + uCyt+	x		x	Aneuploidy of Chromosomes 3, 7 and 17 and loss of chromosome locus 9p21 + protein expression of Carcinoembryonic antigen and Sulphate mucin glycoproteins		FISH + Immunofluorescence		x				AUC	No
3	Base model (age and grade of haematuria) + UroVysion + Cytology			x	Aneuploidy of Chromosomes 3, 7 and 17 and loss of chromosome locus 9p21 + protein expression of Carcinoembryonic antigen and Sulphate mucin glycoproteins		Immunofluorescence		x				AUC	No
4	Optimal (age + FGFR3, TERT, HRAS, ONECUT2 probes 1 + 4, OTX1 probe 2, TWIST)			x	Methylation of ONECUT2 and OXT1 genes and mutation of FGFR3, TERT and HRAS genes	[66]	SNaPshot assay		x	x		Various cut-offs: 0.1213372; 0.1917196; 0.3547327; 0.4975214	SN, SP, PPV, NPV	No
5	Extended Model consisting of: Existing model (univariate analysis incl. age, mutation, methylation) + type of haematuria + gender			x	Mutation of FGFR3, TERT and HRAS and methylation of OTX1, ONECUT2 and TWIST1	[68]	SNaPshot assay and qPCR		x	x			AUC	No
6	Optimal model consisting of: Existing model (univariate analysis incl. age, mutation, methylation) + type of haematuria			x	Mutation of FGFR3, TERT and HRAS and methylation of OTX1, ONECUT2 and TWIST2		SNaPshot assay and qPCR		x	x			SN, SP, PPV, NPV, AUC, FP, FN	No
**BIOMARKER/TESTS, PREDICTION MODELS and CYTOLOGY**														
1	Base model (age, gender, smoker, race, haematuria) + NMP-22	x			Protein expression of Nuclear Matrix Protein 22	[31]	Lateral Flow Test (BladderChek)		x	x			AUC, Accuracy	No
2	Base model (age, gender, smoker, race, haematuria) + NMP-22+ Cytology	x			Protein expression of Nuclear Matrix Protein 22 + Cytology		Lateral Flow Test (BladderChek)		x	x			AUC, Accuracy	No
3	Base model (age, gender, smoker, race, haematuria) + UroVysion + uCyt+ (+) Cytology	x		x	Aneuploidy of Chromosomes 3, 7 and 17 and loss of chromosome locus 9p21 + protein expression of Carcinoembryonic antigen and sulphate mucin glycoproteins + Cytology	[63]	FISH + Immunofluorescence		x				AUC	No
**BIOMARKER/TESTS and OTHER**														
1	CxBladder Triage (CxBT) + Imaging		x		mRNA expression of CDK1, HOXA13, MDK, IGFBP5, CXCR2	[35]	qPCR		x	x		<4.0 to indicate specialist assessment was required	SN, SP, NPV, Accuracy	No
2	Telomerase + Hb Dipstick	x			Telomerase activity and haematuria	[32]	PCR		x				SN, SP	No

SN: Sensitivity; SP: Specificity; PPV: Positive Predictive Value; NVP: Negative Predictive Value; AUC: Area Under Curve; FP: False Positive; FN: False Negative; ^1^ ELISA: Enzyme-linked immunosorbent assay; ^2^ qPCR: Real time Polymerase Chain Reaction; ^3^ RT-PCR: Reverse Transcript Polymerase Chain Reaction; ^4^ ddPCR: droplet digital Polymerase Chain Reaction; ^5^ AuNPs: Gold nanoparticles; ^6^ TRAP: Telomeric Repeat Amplification Protocol; ^7^ NGS: Next Generation Sequencing; ^8^ FISH: Fluorescence In Situ Hybridisation; ^9^ also referred to as ImmunoCyt+.

**Table 3 cancers-15-00709-t003:** Reported Outcomes for Novel Biomarkers by Biomarker Category.

Biomarker Category	Biomarkers/Tests	Studies Identified	Country of Research, Total Population (N) ^1^	Clinical Features				Risk Factors		Short Term Outcomes				Authors’ Conclusions (Applicable in the Health System Where the Biomarker Was Evaluated)
				Haematuria		UTIs	Other	Smoking (N) ^1^	Other (N) ^1^	SN	SP	PPV	NPV	
				VH	NVH									
PROTEINS	ADXBLADDER/Mcm5	Kelly 2012 [51]	UK, 1677	x	x	x	Urolithiasis; other malignancies; other benign conditions	NA	NA	73.0%	68.4%	NA	96.4%	- Useful predictive clinical role e.g., to target imaging and cystoscopic diagnostic procedures for higher risk patients
		Dudderidge 2020 [37]	UK, 856	x	x	x	NA	NA	NA	36.0–85.0% ^2^	47.0–96.0% ^2^	17.0–65.0% ^2^	90.0–96.0% ^2^	- Not affected by benign conditions, such as urinary tract infections or the presence of Haematuria - Easy-to perform ELISA test, compatible with general laboratory equipment available in most hospital settings - Can provide results within 3 h, without the need for a pathologist - Demonstrates potential to replace cytology as an adjunctive test in bladder cancer diagnosis - Cheaper than cytology (no pathologist needed)
	EMA + NMP-52	Attallah 2015 [30]	Egypt, 160	x		NA	Cystitis	NA	NA	94.0%	80.0%	75.0%	95.0%	NA
	uCyt+ + NMP-22	Todenhöfer 2013 [63]	Germany, 808	x		x	Irritative voiding symptoms	NA	NA	90.4%	35.9%	19.0%	95.8%	NA
mRNAs	CxBladder Triage	Davidson 2020 [35]	New Zealand, 884	x	x	x	Cystitis; upper tract stones; vascular prostate; anticoagulation; renal disease; primary amyloidosis	NA	Previous cancer diagnosis *N* = 1; Radiation therapy of pelvis *N =* 1	89.4%	59.0%	NA	98.9%	- Reduces the need for referral to secondary care - Reduces the need for cystoscopies which are invasive - Risk of avoiding a cystoscopy and missing a significant bladder cancer
	CxBladder Triage + Imaging	Davidson2020 [35]								98.1%	98.4%	NA	99.9%	
	Xpert	van Valenberg 2021 [2]	USA, 828	x	x	NA	NA	Current *N* = 139; Former *N* = 288; Never; *N* = 401	Caucasian *N* = 698; Black *N* = 80; Hispanic *N* = 32; Other *N* = 18	78.0%	84.0%	27.0%	98.0% 97.0% ^3^ 99.0% ^4^	- Fast non-invasive method of discriminating BC from other non-serious causes of HM - Reliable, easy to use, fast, and operator independent - Can potentially justly avoid 98% of cystoscopies - Promising tool for identifying Haematuria patients with a low likelihood of BC who might not need to undergo cystoscopy
DNAs	FGFR3 + TERT + HRAS + OXT1 + ONECUT2 + TWIST	van Kessel 2017 [67]	Sweden, Spain, Netherlands, 200	x		NA	NA	NA	NA	93.2%	85.6%	42.2%	99.2%	- Preferred over current clinical practice (i.e., cystoscopy) - Reduces cystoscopies in the primary diagnostic workup - Not feasible in acute Haematuria cases (takes several days to complete) - Availability of urine assay to GPs could expedite referrals - In primary setting may add value as could reduce diagnostic delay for female patients with Haematuria
		van Kessel 2020 [68]	Netherlands, 838	x	x	NA	NA	Current *N* = 221; Former *N* = 185; Never *N* = 258; Not reported *N* = 174	NA	92.0%	73.0%	34.0%	98.0%	
PROTEIN + DNA	UroVysion + NMP-22 + uCyt+	Todenhöfer 2013 [63]	Germany, 808	x		x	Irritative voiding symptoms	NA	NA	83.5%	74.1%	34.9%	96.4%	- Relatively high costs of performing multiple urine tests

VH: Visible Haematuria; NVH: non-visible Haematuria; UTIs: Urinary Tract Infections; N: (number); SN: sensitivity; SP: specificity; PPV: positive predictive value; NPV: negative predictive value. ^1^ Numbers reported where available. ^2^ Range reported to include all different cut-off points used for testing. ^3^ In patients with non-visible Haematuria. ^4^ In patients with visible Haematuria.

## Data Availability

The Search Strategy used in this systematic review has been provided in Appendix A. The datasets generated during the study are available from the corresponding author on reasonable request.

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
