# Peer review of "Diagnostic Performance of Biomarkers for Bladder Cancer Detection Suitable for Community and Primary Care Settings: A Systematic Review and Meta-Analysis"

_cancers, 2023, doi:10.3390/cancers15030709_

Round 1

Reviewer 1 Report

The author has reported Diagnostic performance of biomarkers for bladder cancer de- 2 tection suitable for community and primary care settings: a sys- 3 tematic review and meta-analysis. It is well written and have excellent presentation. I recommended it for publication after minor changes.

1.  What kind of biomarker? please make clear for reader.

2. Is only CT useful for diagnosis?

3.  please update reference with following 

  • DOI: 
  • 10.1021/acsomega.2c05948.  
    • DOI: 
    • 10.52700/pjbb.v3i1.107
  • DOI: 
  • 10.1021/acs.bioconjchem.1c00437
  •    
  • 4. what are main challenges in diagnosis filed, please highloght it in the end. 
  •  

Author Response

We would like to thank Reviewer 1 for their insightful comments and suggestions. Please find below our response to each point raised highlighted in red.

Point 1: What kind of biomarker? please make clear for reader.

Response 1: Based on the inclusion/exclusion criteria pertaining to Intervention (see Appendix 2. PICOS Framework - Inclusion & Exclusion Criteria in Supplementary Materials), this systematic review aimed to include all biomarkers irrespective of sample source “biomarkers derived from human blood (serum, plasma), urine, faecal, salivary or breath samples”. Similarly, we aimed to include biomarkers (individual, panel, and/or combinations) of any category type (proteins, DNAs, and mRNAs). Therefore, the kind of biomarker (either in terms of category or sample source) was purposively not further specified in text. The only specification was that the sample source was non-invasive so samples could be collected in community and primary care settings (in line with the review aim to identify biomarkers suitable for use in these settings). We have added a sentence to our methods section (2.2. Inclusion & Exclusion Criteria, end of first paragraph) to emphasise this:

“We were interested in any biomarker feasible to use in community and primary care settings, i.e., identified in non-invasive samples such as blood (serum or plasma), urine, faeces, saliva or breath.”

In our results section (3.5. Biomarker Characteristics), we specified the range of biomarkers in terms of biomarker category and sample source (predominantly urine but also urine and plasma) (see below).

“In terms of category, 52 biomarkers were classified as proteins (including single proteins, combinations of proteins and combinations of proteins with prediction models, cytology, and other tests), 36 as  DNAs and 18 as mRNAs (all following the same pattern as in proteins). There were also nine biomarkers combining proteins and mRNAs and six biomarkers combining proteins and DNAs. The discrepancy in the total number of biomarkers per category (N=121) and the total number of biomarkers reported (N=112) is due to different biomarker categories pertaining to the same biomarker being reported together (an example of this is Telomerase in Table 2). All biomarkers were sampled from urine - apart from one (CYFRA21-1) which was sampled from both urine and serum using a range of test platforms such as enzyme-linked immunoassay (ELISA), fluorescence in situ hybridisation (FISH), Lateral Flow Test, and different types of polymerase chain reaction (PCR).”

Point 2: Is only CT useful for diagnosis?

Response 2: In our Introduction (paragraph 2), we gave an overview of the typical diagnostic pathway for bladder cancer, which involves a range of investigations. These include primary care tests as well as specialist investigations including imaging tests such as CT: (see below)

“The diagnostic pathway for bladder cancer involves a combination of investigations, from urine tests in the community and primary care to specialist investigations such as upper urinary tract imaging (including ultrasound and Computed Tomography (CT) scans), urine cytology and cystoscopy. The latter remains the gold standard for bladder cancer detection in patients investigated following haematuria [6-7].”

Point 3: Please update reference with following 

  • DOI: 10.1021/acsomega.2c05948.  
  • DOI: 10.52700/pjbb.v3i1.107
  • DOI: 10.1021/acs.bioconjchem.1c0043

Response 3: We have carefully reviewed the references provided and while interesting in terms of the evidence presented, the team felt that they do not seem to fit into the focused aim of our systematic review, which was to identify biomarkers for bladder cancer that may be suitable for use in community and primary care settings. Since this systematic review, however, is part of ongoing work on cancer detection with the CanTest Collaborative, we would like to thank the reviewer for sharing these sources of evidence that may be helpful in future work.

Point 4: What are main challenges in diagnosis filed, please highlight it in the end. 

Response 4: We have interpreted this comment as the need to highlight: 1) the existing challenges in diagnosing bladder cancer in the general population (which is the focus of our review), and 2) the challenges in the use of biomarkers in the diagnosis of bladder cancer. We have highlighted these issues in two sections of the paper respectively:

  1. In our Introduction, we explained the current limitations and challenges with the current diagnostic process and tests, and therefore the need for adjunct diagnostic tools:

“Patients presenting with non-visible haematuria and other urological symptoms, such as lower urinary tract symptoms, may cause diagnostic challenges. This is because these symptoms are common in the general population and are more likely to be due to benign causes rather than cancer (Zhou et al. 2019). Identifying tools to improve the diagnostic pathway may improve diagnostic timeliness, and therefore outcomes, for patients with bladder cancer.” [Paragraph 1]

“The diagnostic pathway for bladder cancer involves a combination of investigations, from urine tests in the community to specialist investigations such as upper urinary tract imaging (including ultrasound and Computed Tomography (CT) scans), urine cytology and cystoscopy. The latter remains the gold standard for bladder cancer detection in patients investigated following haematuria (Tan et al. 2018; Ng et al. 2021). Disadvantages of these tests, such as poor sensitivity of ultrasound, radiation exposure associated with CT, and invasiveness of cystoscopy, can limit their use in the general population (Zhu et al. 2019). There is, therefore, an urgent need to identify new approaches for improving risk stratification of symptomatic patients to improve early detection of bladder cancer and reduce burden of unnecessary investigations for patients.” [Paragraph 2]

  1. In our Discussion (4. Implications for research and practice), we explained that although some biomarkers may be suitable for use in primary care, there are further challenges that need to be overcome to enable their use. We feel that this section adequately addressed many challenges relating to the potential use of the identified biomarkers. Some examples of these are in the Implications for research and practice section (as listed below):

“Similar to conclusions from previous reviews (Hong et al. 2021; Sciarra et al. 2021) while there are promising results (particularly regarding high NPVs) for some biomarkers, additional validations are still needed in the community setting.” [Paragraph 1]

“It is also important to consider the role of the biomarker within the cancer diagnostic pathway. (…) We found no studies reporting the use of biomarkers for bladder cancer in this context. To assess the clinical utility of these biomarkers in the community, there is therefore a need to evaluate these biomarkers in the general population, at the pre-referral stage of the diagnostic process.” [Paragraph 2]

“Novel biomarkers showing promising results need to be further evaluated, preferably prospectively, with consistency regarding populations, care settings and thresholds/cut-off points used.” [Paragraph 3]

We would be happy to further expand should the Editors feel necessary, or if we have misunderstood the Reviewer’s comment.

Reviewer 2 Report

Papavasiliou E. et al., have discussed “Diagnostic performance of biomarkers for bladder cancer suitable for community and primary care settings: a systematic review”. It seems quite promising in the current time. The way of presentation of the systematic and metaanalysis looks fine. There are some concerns:

1.    I do not agree with “2.7. Data Synthesis”. it cannot be data synthesis

2.    I also see FGFR3 while there are more sound literatures for FGFR1,2, & 4. There are only few selective references for FGFR3.

3.    The authors should look for a number typos issues.

Author Response

We would like to thank Reviewer 2 for their insightful comments and suggestions. Please find below our response to each point raised highlighted in red.

Point 1: I do not agree with “2.7. Data Synthesis”. it cannot be data synthesis

Response 1: Thank you for this helpful comment. This has been now changed to “2.7. Data Analysis”.

Point 2:   I also see FGFR3 while there are more sound literatures for FGFR1,2, & 4. There are only few selective references for FGFR3.

Response 2: Thank you for your observation. In our systematic review, we have followed a strict study selection process which was implemented in a systematic way. Included studies had to be situated within Phase 2 (i.e., providing measures of diagnostic accuracy beyond discovery/development, even if tested in high prevalence populations) and Phase 3 (i.e., examining diagnostic accuracy in intended low prevalence populations, and providing measures of clinical utility including feasibility and acceptability) of the CanTest framework (Walter et al. 2019). We also only included patients with cancer suspicion at the point of recruitment (therefore excluding studies recruiting populations already diagnosed with cancer). This was important to reduce the risk of spectrum bias (different test performance in different populations) and resulted in the exclusion of many ineligible studies.

All biomarkers identified in included studies were reported. While there might be literature for FGFR1, 2, & 4, it could not be added to this review as these biomarkers were not identified in any of the studies that met our inclusion criteria (described in the method section and Appendix 2).

Point 3: The authors should look for a number of typos issues.

Response 3: The manuscript has been scrutinised and any typos identified have been addressed.

Reviewer 3 Report

Hello Authors,

Thank you all for your hard work and dedication for making this amazing article in improving diagnostic performance of bladder cancer and finding novel biomarkers. Your efforts would really benefit this journal and its audience. 

After reviewing your article, I feel the only change at this time would be to update your identification and eligibility criteria to have more broad scope of record to be included in the study. Initially screen provided you 6635 records but as following the PRISMA chat I see only 44 where used. This number seems to be very low to find any concrete conclusion. If you increase your search criteria you can find more records, which could lead to finding more novel biomarkers. 

Author Response

We would like to thank Reviewer 3 for their insightful comment and suggestion. Please find below our response highlighted in red.

Point 1: Thank you all for your hard work and dedication for making this amazing article in improving diagnostic performance of bladder cancer and finding novel biomarkers. Your efforts would really benefit this journal and its audience. 

After reviewing your article, I feel the only change at this time would be to update your identification and eligibility criteria to have more broad scope of record to be included in the study. Initially screen provided you 6635 records but as following the PRISMA chat I see only 44 where used. This number seems to be very low to find any concrete conclusion. If you increase your search criteria you can find more records, which could lead to finding more novel biomarkers. 

Response 1: Thank you for the positive comments on the quality of our paper. While there is indeed vast literature on biomarkers for bladder cancer, we were only interested in those that could be suitable for use in community and primary care settings. This review is therefore different from broader systematic reviews conducted to date, but this was intentional in order to meet our aim. We wanted to identify novel biomarkers for bladder cancer detection that might be suitable for use in the general population in community and primary care settings, often the first point of contact for patients in the healthcare system. To ensure this aim was successfully addressed, we developed and implemented a rigorous two-fold approach to study selection using the PICOS Framework (Population, Intervention, Comparators/Context, Outcomes and Study) (see Appendix 2. PICOS Framework - Inclusion & Exclusion Criteria in Supplementary Materials) and the CanTest framework (see Appendix 3: CanTest Framework in Supplementary Materials). Based on the former, a list of strict inclusion/exclusion criteria was created to account for study selection. Based on the latter, only studies situated within Phase 2 (i.e., providing measures of diagnostic accuracy beyond discovery/development, even if tested in high prevalence populations) and Phase 3 (i.e., examining diagnostic accuracy in intended low prevalence populations, and providing measures of clinical utility including feasibility and acceptability) were eligible for inclusion in our review. Even when searches were updated, it was this scrutiny in our study selection approach that resulted in the low number of included studies. This was important to ensure relevance of included studies and meet the review aims.

Reviewer 4 Report

The authors conducted a systematic review and meta-analysis on the diagnostic performance of biomarkers for bladder cancer detection that are appropriate for use in community and primary care settings. The manuscript is well written. I only have a few minor comments.

1. The tables included in this manuscript are not formatted properly (certain columns are outside of the page border).

2. Can the authors please provider higher resolution images for Figure 3? 

Author Response

We would like to thank Reviewer 4 for their comments and suggestions. Please find below our response to each point raised highlighted in red.

Point 1: The tables included in this manuscript are not formatted properly (certain columns are outside of the page border).

Response 1: This is due to the size of tables that were originally submitted in A3 page layout. When files were merged during submission, however, page layout seems to have changed to A4 which caused the formatting errors (with certain columns of the tables included in the manuscript being outside the page border). We have created a new version of the tables converting from PDF to JPEG that allows for them to fit into A4 page layout for the purpose of this resubmission. We would appreciate any input from the journal editor on how to proceed if the paper is accepted for publication.

Point 2: Can the authors please provide higher resolution images for Figure 3? 

Response 2: A new version of Figure 3 (of higher resolution) has been included.